# PERFT: PARAMETER-EFFICIENT ROUTED FINE-TUNING FOR MIXTURE-OF-EXPERT LARGE LANGUAGE MODEL

## ABSTRACT

The Mixture-of-Experts (MoE) paradigm has emerged as a promising approach for scaling transformer-based large language models (LLMs) with improved resource utilization. However, efficiently fine-tuning MoE LLMs remains largely underexplored. Inspired by recent works on Parameter-Efficient Fine-Tuning (PEFT), we present a unified framework for integrating PEFT modules into MoE LLMs. Our framework, aligned with the core mechanisms of MoE, encompasses a comprehensive set of design dimensions including various functional and composition strategies. By combining the key design choices within our framework, we introduce **P**arameter-**E**fficient **R**outed **F**ine-**T**uning (**PERFT**) as a flexible and scalable family of PEFT strategies tailored for MoE LLMs.[1] Extensive experiments adapting OLMoE-1B-7B and Mixtral-8×7B for various commonsense and arithmetic reasoning tasks demonstrate the effectiveness, scalability, and intriguing dynamics of PERFT. Additionally, we provide empirical findings for each specific design choice to facilitate better application of MoE and PEFT.

## 1 INTRODUCTION

As modern transformer-based Vaswani et al. (2017) large language models (LLMs) continue to scale up, Mixture-of-Experts (MoE) (Shazeer et al., 2017) has emerged in recent years as a promising solution to the trade-off between performance and cost, yielding notable results in a series of frontier models (Jiang et al., 2024; Reid et al., 2024; Dai et al., 2024; Qwen, 2024; Grok, 2024). With so many new MoE LLMs available, how to effectively fine-tune them for downstream tasks has become an area of considerable value. The advancements of MoE do not directly translate to efficiency in their fine-tuning, and full fine-tuning these models remains prohibitively expensive due to their immense number of expert parameters. Besides, the routing mechanism among sparsely-activated experts poses unique challenges unseen in conventional dense architectures (Wang et al., 2024). This necessitates exploring solutions specially-designed for efficiently adapting sparse MoE models, without incurring the full cost of fine-tuning all parameters.

Parameter-Efficient Fine-Tuning (PEFT) techniques, such as adapters (Houlsby et al., 2019) and LoRA (low-rank adaptation; Hu et al., 2022), have gained considerable attention on conventional dense models. Combining hybrid elements from different PEFT methods have also shown promising results (He et al., 2022; Hu et al., 2023; Zhang et al., 2023). With the rise of MoE architectures, recent studies have explored PEFT solutions for dense models with MoE-inspired designs (Zadouri et al., 2023; Dou et al., 2023; Luo et al., 2024; Li et al., 2024; Gao et al., 2024; Wu et al., 2024). However, designing PEFT strategies tailored for MoE models remains largely underexplored.

To this end, we present the first unified framework for incorporating diverse PEFT modules directly into the MoE mechanism. Different from previous PEFT solutions that operate in isolation from the underlying MoE architecture, our framework is designed closely around the unique routing mechanisms among experts in MoE models. We introduce two key design dimensions. **Functional strategies** define the internal mechanisms of the introduced PEFT module, including the architecture inside individual PEFT modules, the multiplicity of PEFT modules, and the routing mechanism among them. **Compositional strategies** describe how PEFT modules interact with the original MoE

---

[1] Code available via `https://anonymous.4open.science/r/PERFT-MoE/`.

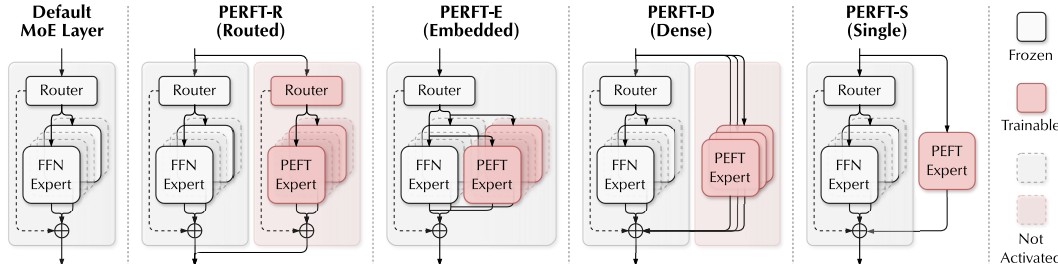

Figure 1: **Illustration of a default MoE layer and the PERFT family.** PERFT-R, the primary variant, holds an independent routing among the introduced PEFT experts. PERFT-E embeds PEFT experts within the original MoE module and directly utilizes its routing patterns. PERFT-D and PERFT-S simply work as independent shared expert(s) alongside the MoE module.

architecture, including operating as shared PEFT experts or embedded PEFT experts. To rigorously characterize the behavior of adapting MoE LLMs with each strategies, we provide empirical analyses that offer insights into understanding and optimizing configurations on these dimensions.

By exploring representative design choices within our framework, we introduce **P**arameter-**E**fficient **R**outed **F**ine-**T**uning (**PERFT**), a flexible and scalable family of PEFT strategies tailored for MoE LLMs, as shown in Figure 1. These methods cover a range of architectural designs with varying levels of scale, sparsity, and routing dynamics. At the core of PERFT is PERFT-R (Routed), which introduces an independent routing mechanism among multiple PEFT experts, enabling task-specific expert activation patterns. We also study PERFT-E (Embedded), which utilizes the pre-trained router, and PERFT-D (Dense) and PERFT-S (Single), which employ always-activated PEFT experts without routing. These variants cover a wide range of functional and compositional strategies, allowing for a systematic exploration on the trade-offs between parameter efficiency, sparsity, and routing in fine-tuning MoE modules.

Extensive experiments are conducted on OLMoE-1B-7B (Muennighoff et al., 2024) and Mixtral-8×7B (Jiang et al., 2024) for commonsense and math reasoning tasks. Our results demonstrate that PERFT enables different levels of efficient adaptation of MoE LLMs while maintaining competitive performance. With an equivalent level of activated trainable parameters in OLMoE-1B-7B, PERFT-R achieves improvements of up to 17.2% and 12.3% over the average performance of MoE-agnostic baseline methods in each domain. We also demonstrate and empirically analyze our observations for the optimal scaling, sparsity, and routing configurations that generalize across settings. We hope to provide practical insights for improving future MoE and PEFT approaches, and contribute to the understanding of adaptation strategies for modern large-scale LLMs.

The primary contributions of our work are as follows:

1. We introduce a unified framework of PEFT techniques tailored for MoE LLMs. This encompasses multiple dimensions of design strategies, offering a novel perspective.
2. By combining the design choices within this unified framework, we propose PERFT as a flexible and scalable family of strategies for adapting MoE LLMs.
3. Extensive experiments adapting OLMoE-1B-7B and Mixtral-8×7B for commonsense and arithmetic reasoning tasks validate the effectiveness, scalability, and intriguing dynamics of PERFT. We provide empirical findings and analysis for each specific design choice.

## 2 BACKGROUND

### 2.1 MIXTURE-OF-EXPERTS IN TRANSFORMER MODEL

**Transformer Model.** Consider a transformer model comprising $L$ layers of transformer blocks, each incorporating a standard self-attention mechanism and a feed-forward neural network (FFN). Given a sequence of $T$ tokens with an initial embedding in a $D$-dimensional hidden space $\boldsymbol{x}_0^{1:T} \in \mathbb{R}^{T \times D}$, we formulate the inner mechanism of each transformer block[2] at layer $l \in \{1, \cdots, L\}$ as:

$$\boldsymbol{h}_l^{1:T} = \texttt{SelfAttn}_l\left(\boldsymbol{x}_{l-1}^{1:T}\right) + \boldsymbol{x}_{l-1}^{1:T}, \quad \boldsymbol{x}_l^t = \texttt{FFN}_l\left(\boldsymbol{h}_l^t\right) + \boldsymbol{h}_l^t, \tag{1}$$

---

[2]Layer normalization and dropout operations are omitted in this paper for clarity.

where $\boldsymbol{h}_l^{1:T}$ denotes the attention module output with the residual connection. The Feed-Forward Network $\texttt{FFN}_l$ performs a token-wise mapping, yielding output $\boldsymbol{x}_l^t$ at token $t \in \{1, \cdots, T\}$ with residual added, which subsequently becomes the input for the next transformer block at layer $l + 1$.

**Mixture-of-Experts.** As a viable solution to the computational challenges in scaling models and improving specialization, early forms of MoE were introduced (Jacobs et al., 1991; Jordan & Jacobs, 1994; Eigen et al., 2013; Shazeer et al., 2017). In the era of transformers, studies have revealed that FFNs, with two-thirds of the model parameter, encapsulate a substantial amount of knowledge (Geva et al., 2021; Dai et al., 2022) that can be attributed to sparsely represented features (Dalvi et al., 2019; Durrani et al., 2020; Gurnee et al., 2023). Leveraging this internal sparsity, MoE architectures can achieve better resource utilization by activating only a subset of effective parameters for each input (Liu et al., 2023b), which has since been successfully applied to transformer-based language models (Lepikhin et al., 2020; Du et al., 2022; Fedus et al., 2022; Zoph et al., 2022a; Komatsuzaki et al., 2022; Rajbhandari et al., 2022; Jiang et al., 2024; Dai et al., 2024; Qwen, 2024; Grok, 2024). Modern MoE architectures employ token-wise gating network (router) $G(\cdot)$, which dynamically assigns each token to $K$ of top-activated experts among $N$ FFN experts $E_i(\cdot)$:

$$\texttt{MoE}(\boldsymbol{h}^t) = \sum_{i=1}^{N} \left( G\left(\boldsymbol{h}^t\right)_i E_i\left(\boldsymbol{h}^t\right) \right), \quad \text{where } G\left(\boldsymbol{h}^t\right) = \texttt{TopK}\left(\texttt{Softmax}\left(\boldsymbol{h}^t \boldsymbol{W}_g\right), K\right), \quad (2)$$

in which $G(\cdot) : \mathbb{R}^D \mapsto \mathbb{R}^N$ denotes the sparse gating function that distributes weights across all $N$ FFN experts' outputs, among which only $K$ get nonzero values. The weight matrix $\boldsymbol{W}_g$ in $G(\cdot)$ can be interpreted as a set of $D$-dimensional column vectors $\{\boldsymbol{g}_i | i \in 1, \cdots, N\}$, each corresponding to a characteristic hidden state $\boldsymbol{h}_i$ for the expert $E_i$. The router computes token-to-expert affinity scores $\boldsymbol{s}_i^t$ via a softmax-normalized projection of each token's hidden state onto these characteristic states (Zhou et al., 2022; Dikkala et al., 2023; Lo et al., 2024), which are subsequently top-K thresholded to yield expert selection results for each token. Notably, recent works (Gou et al., 2023; Dai et al., 2024; Qwen, 2024) have explored *shared experts* that structurally mirror routed experts, working in parallel with them and always remaining activated for capturing common knowledge.

## 2.2 PARAMETER-EFFICIENT FINE-TUNING FOR TRANSFORMER-BASED MODEL

**Vanilla PEFT.** Classical full fine-tuning approaches for downstream tasks (Devlin et al., 2019; Qiu et al., 2020) have become increasingly impractical as transformers continue scaling up. Recent work has introduced diverse PEFT methods offering comparable performance to full fine-tuning with significantly reduced computational demands. He et al. (2022) present a unified view for PEFT, where any PEFT method can be viewed as a combination of several design dimensions. For instance, given the adapted module's input $\boldsymbol{h}$ and output $\boldsymbol{x}$, LoRA (Hu et al., 2022), which approximates weight updates using low-rank matrices, can be described as a parallel operation $\Delta(\boldsymbol{h}) = \boldsymbol{h} \boldsymbol{W}_{\text{down}} \boldsymbol{W}_{\text{up}}$ and $\boldsymbol{x} \leftarrow \boldsymbol{x} + s \cdot \Delta(\boldsymbol{h})$. This framework facilitates hybrid design for better PEFT variants. They find that parallel PEFT modules generally outperform sequential adaptations, and modifying FFN yields better results than modifying attention, which are further supported by Hu et al. (2023), Zhang et al. (2023), Dettmers et al. (2024) and Hao et al. (2024).

**PEFT with MoE-like Structures.** The success of MoE transformers has inspired MoE-structured adaptations. Much recent work has focused on developing such modules for dense models, including inserting multiple LoRA experts with routers at attention layers (Liu et al., 2023a; Luo et al., 2024) and alongside dense FFN layer (Zadouri et al., 2023; Dou et al., 2023; Page-Caccia et al., 2024; Chen et al., 2024; Hao et al., 2024). Gao et al. (2024) find that allocating more LoRA experts to higher layers leads to better performance. Li et al. (2024) propose up-cycled a mixture of LoRA-adapted frozen FFN experts from dense models. Wu et al. (2024) explore methods for composing multiple trained LoRAs in a MoE style. Notably, all these methods primarily focus on adapting dense models, leaving the application of PEFT to inherently sparse MoE models largely underexplored. Recently Wang et al. (2024) propose an expert-specialized fine-tuning approach, which comes closest to this research gap by selectively fine-tuning the most relevant experts for downstream tasks, though no PEFT techniques are involved. Our work, in contrast, directly addresses this area by introducing PEFT modules into the MoE mechanism, which offers a more flexible and efficient solution for adapting MoE models while preserving their original weights untouched.

## 3 METHODOLOGY

### 3.1 THE UNIFIED FRAMEWORK

This section introduces our unified framework for PEFT on MoE models. Inspired by the unified view of PEFT (He et al., 2022), our framework focuses on two key design dimensions, as shown in Figure 2. **Functional strategies** define the internal mechanism of the introduced PEFT module, including the architecture inside individual PEFT modules, the multiplicity of PEFT modules, and the routing mechanisms among them. **Compositional strategies** describe how PEFT modules interact with the original MoE architecture, including operating as shared PEFT experts or embedded PEFT experts. By considering these aspects, our framework addresses the unique mechanisms of both PEFT and MoE, providing a novel and comprehensive perspective on adapting MoE LLMs.

### 3.1.1 FUNCTIONAL STRATEGY

This dimension describes the internal implementation of the introduced PEFT module. We consider variations of mechanisms in three dimensions:

**Architecture inside PEFT Experts.** This aspect defines the specific internal structure of each individual PEFT expert. The general architecture for computing $\Delta(\boldsymbol{h})$ in each PEFT expert can be formalized as

$$\Delta(\boldsymbol{h}) = \texttt{UpProj}(\texttt{Act}(\texttt{DownProj}(\boldsymbol{h}))), \quad (3)$$

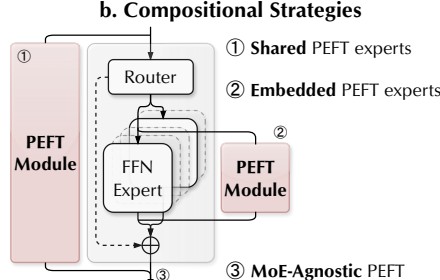

Figure 2: **The unified framework of PEFT for a MoE module. a.** Functional strategies specify the internal implementation of the introduced PEFT module. **b.** Compositional strategies describe the PEFT module's interaction with the original MoE mechanism.

where $\texttt{Act}(\cdot)$ is implemented with non-linear activation functions, or with an identity function for LoRA. The $\texttt{DownProj}(\cdot) : \mathbb{R}^D \mapsto \mathbb{R}^{D_B}$ and $\texttt{UpProj}(\cdot) : \mathbb{R}^B \mapsto \mathbb{R}^{D_B}$ introduce a key scaling factor, the *bottleneck* size $D_B$, known as *rank* $r$ used in LoRA's low-rank decomposition. Adjusting $D_B$ leads to linear scaling of trainable parameters. Optimizing this hyperparameter is crucial for different tasks and models, as it balances the bottleneck subspaces' capacity for additional knowledge against the effectiveness of training newly introduced weights with given data (Hu et al., 2022).

**Multiplicity of PEFT Experts.** The number of PEFT experts serves as another key scaling factor in our framework. Increasing the number of PEFT experts allows each to generate its own copy of $\Delta(\boldsymbol{h})$, denoted as $\Delta_i(\boldsymbol{h})$. Previous studies on fine-tuning dense models with MoE-like structures (Zadouri et al., 2023; Liu et al., 2023a; Dou et al., 2023; Li et al., 2024) have empirically shown that optimizing the number of adapters can significantly impact performance. This optimization can be tailored to specific tasks, models, or even individual layers within a model (Gao et al., 2024). We investigate the balance between performance and effective utilization of experts in our experiments.

**Routing among PEFT Experts.** This aspect considers whether an independent routing mechanism is introduced among PEFT experts. In contrast to previous work primarily focusing on adapting dense models using PEFT modules with MoE-like structures (Hao et al., 2024; Gao et al., 2024; Wu et al., 2024), our framework reveals the potential dynamics in the interaction between routed PEFT experts and the pretrained MoE module. For a token-wise routing among $M$ PEFT experts, the PEFT module operates similarly to the original MoE module for FFN experts (Equation 2):

$$\Delta(\boldsymbol{h}^t) = \sum_{i=1}^{M} \left( \tilde{G}\left(\boldsymbol{h}^t\right)_i \Delta_i(\boldsymbol{h}^t) \right), \quad (4)$$

where $\tilde{G}(\cdot)$ denotes the gating function for the PEFT experts. This aspect highlights the profound dynamics between routers and experts in MoE and PEFT modules, as shown in Figure 3. Based on the key-value memory perspective for FFN (Geva et al., 2021) (Figure 3a), we can similarly interpret the weight matrix $\boldsymbol{W}_g \in \mathbb{R}^D \times \mathbb{R}^N$ in a router for $N$ FFN experts as a set of $N$ individual vectors $\{\boldsymbol{g}_i\}$, each representing a characteristic hidden state for the corresponding expert's key memories. More specifically, each of the $N$ vectors approximately symbolizes a cluster of all individual neuron vectors within each FFN expert, and the routing process can be interpreted as a projection of the current hidden state onto these $N$ vectors to calculate the affinity of each expert with the input token. For our PEFT expert router $\tilde{G}(\cdot)$, we can either learn from scratch a new collection of PEFT

Figure 3: **The dynamics between key memory vectors in experts and expert vectors in routers.**
**a.** A dense FFN expert as projecting $\boldsymbol{h}^t \in \mathbb{R}^D$ onto $D_a$ key memory vectors in the weight matrix $\boldsymbol{W}_{\text{up}} = \{\boldsymbol{k}_i \in \mathbb{R}^D\}$ and yielding activation scores $\boldsymbol{a}^t \in \mathbb{R}^{D_a}$ distributed over the key memories.
**b.** A router for $N$ FFN experts as projecting $\boldsymbol{h}^t$ onto $N$ expert vectors stored in router weight matrix $\boldsymbol{W}_g = \{\boldsymbol{g}_i \in \mathbb{R}^D\}$, yielding token-to-expert affinity scores $\boldsymbol{s}^t \in \mathbb{R}^N$ distributed over the experts. Each expert vector $\boldsymbol{g}_i$ symbolizes a characteristic $\boldsymbol{h}^t$ pattern featuring its expert's key memory vectors $\{\boldsymbol{k}_j\}_i$. **c.** Routers for both the $N$ FFN experts and $M$ PEFT experts introduce interesting dynamics between their expert vectors $\{\boldsymbol{g}_i\}$ and $\{\tilde{\boldsymbol{g}}_i\}$, resulting a more flexible space for fine-tuning.

expert vectors $\{\tilde{\boldsymbol{g}}_i\}$, or directly utilize the existing $\{\boldsymbol{g}_i\}$ from the original router for FFN experts, which becomes functionally equivalent to the configuration of embedded PEFT in Section 3.1.2. We provide detailed visualization and analysis of these dynamics in our experiments.

### 3.1.2 COMPOSITIONAL STRATEGY

The compositional strategy defines how the PEFT module integrates with the original MoE model. Based on findings from previous research (He et al., 2022; Hu et al., 2023; Luo et al., 2024; Hao et al., 2024) that inserting PEFT modules in parallel generally yields superior performance, we focus exclusively on *parallel* insertion methods, i.e., PEFT receiving the same input as the module it is adapting and combining its output with that of the same module. This consideration aligns with the parallel nature of MoE architectures, where FFN experts operate concurrently rather than in a stacked configuration. Here we identify three main categories of insertion strategies:

**Shared PEFT Experts.** The PEFT module can operate in parallel with the entire MoE module, functioning as shared PEFT experts. Given a input hidden state sequence $\boldsymbol{h}^{1:T}$, we have:

$$\boldsymbol{x}^{1:T} = \sum\nolimits_{i=1}^{N} \left( G\left(\boldsymbol{h}^{1:T}\right)_i E_i\left(\boldsymbol{h}^{1:T}\right) \right) + \Delta(\boldsymbol{h}^{1:T}) + \boldsymbol{h}^{1:T}, \tag{5}$$

where the PEFT module takes the same input $\boldsymbol{h}^{1:T}$ as the MoE module, and combines its output additively with the MoE output to the residual connection. This approach draws inspiration from the concept of shared FFN experts in recent works (Gou et al., 2023; Dai et al., 2024; Qwen, 2024). Introducing these shared structurally identical FFN experts alongside routed FFN experts during training MoE models aims to improve parameter efficiency by mitigating the redundancy of shared knowledge across routed experts. Applying this principle to lightweight PEFT modules, we hypothesize that these shared PEFT experts can similarly capture and adapt the common parts needed among routed FFN experts, thereby potentially offering greater efficiency as well.

**Embedded PEFT Experts.** In this configuration, the PEFT modules are embedded within the MoE module. Each PEFT module is paired with a corresponding FFN expert and operates in a tight coupling manner, receiving the same token-wise input $\boldsymbol{h}^t$ as distributed by the MoE router:

$$\boldsymbol{x}^t = \sum\nolimits_{i=1}^{N} G(\boldsymbol{h}^t)_i \left( E_i(\boldsymbol{h}^t) + \Delta_i(\boldsymbol{h}^t) \right) + \boldsymbol{h}^t, \tag{6}$$

where $E_i(\boldsymbol{h}^t)$ is the output of the $i$-th FFN expert for token $t$, and $\Delta_i(\boldsymbol{h}^t)$ is the output for token $t$ of the $i$-th PEFT module that is associated with the $i$-th expert. The PEFT modules' outputs are combined with their corresponding FFN experts' outputs before being weighted by the router and summed. This formulation can be viewed as introducing $N$ PEFT experts embedded within the MoE module, mirroring the activation patterns of the original FFN experts as discussed in Section 3.1.1.

**MoE-Agnostic PEFT.** The PEFT module is integrated at locations independent of the MoE modules, completely decoupled and functioning agnostically to the MoE mechanism. This includes

previous PEFT strategies that treat models effectively as if they were dense architecture. We include this strategy as a baseline in our experiments, enabling us to compare the performance of trivial techniques applied without consideration of the underlying MoE structure.

## 3.2 THE PERFT FAMILY

Deriving from our unified framework of PEFT on MoE LLMs, we hereby propose **P**arameter **E**fficient **R**outed **F**ine-**T**uning (PERFT) as a family of novel PEFT methods tailored for MoE models, as illustrated in Figure 1. At the core of the PERFT family is **PERFT-R (Routed)**, with a parallel module consisting of an independent router among the introduced PEFT experts:

$$\boldsymbol{x}^{1:T} = \sum_{i=1}^{N} \left( G\left(\boldsymbol{h}^{1:T}\right)_i E_i\left(\boldsymbol{h}^{1:T}\right) \right) + \sum_{j=1}^{M} \left( \tilde{G}\left(\boldsymbol{h}^{1:T}\right)_j \Delta_j\left(\boldsymbol{h}^{1:T}\right) \right) + \boldsymbol{h}^{1:T}, \qquad (7)$$

where $\tilde{G}(\cdot) : \mathbb{R}^D \mapsto \mathbb{R}^M$ denotes the gating function for the $M$ PEFT experts $\Delta_j(\cdot)$. PERFT-R allows for learning an independent series of expert vectors $\tilde{\boldsymbol{g}}_i$ for PEFT experts, together with FFN expert vectors $\boldsymbol{g}_i$ forming an intriguing dynamics, as discussed in Section 3.1.1 and Figure 3c.

If the number of introduced PEFT experts $M$ matches the number of FFN experts $N$ in the original MoE module, the structural design in PERFT-R provides a possibility to substitute $\tilde{G}(\cdot)$ with the original $G(\cdot)$, which makes it becomes a simplified special case

$$\begin{aligned} \boldsymbol{x}^{1:T} &= \sum_{i=1}^{N} \left( G\left(\boldsymbol{h}^{1:T}\right)_i E_i\left(\boldsymbol{h}^{1:T}\right) \right) + \sum_{j=1}^{N} \left( G\left(\boldsymbol{h}^{1:T}\right)_j \Delta_j\left(\boldsymbol{h}^{1:T}\right) \right) + \boldsymbol{h}^{1:T} \\ &= \sum_{i=1}^{N} G\left(\boldsymbol{h}^{1:T}\right)_i \left( E_i\left(\boldsymbol{h}^{1:T}\right) + \Delta_j\left(\boldsymbol{h}^{1:T}\right) \right) + \boldsymbol{h}^{1:T}, \end{aligned} \qquad (8)$$

which takes exactly the same form as the embedded PEFT experts in Equation 6. Hence we denote this variant as **PERFT-E (Embedded)**. As it directly utilizes the expert vectors $\boldsymbol{g}_i$ original pretrained router for distributing tokens for PEFT experts instead of learning weights from scratch, it can be intuitively estimated that this property of would lead to performance gain especially when the number of routed experts are to some extent that learning from scratch is not able to capture enough quality distribution of PEFT expert vectors in the space of hidden states.

By removing routing functions and naively making multiple PEFT shared experts always activated in parallel with the MoE module, we have another variant **PERFT-D (Dense)**, denoted as

$$\boldsymbol{x}^{1:T} = \sum_{i=1}^{N} \left( G\left(\boldsymbol{h}^{1:T}\right)_i E_i\left(\boldsymbol{h}^{1:T}\right) \right) + \sum_{j=1}^{M} \Delta_j\left(\boldsymbol{h}^{1:T}\right) + \boldsymbol{h}^{1:T}, \qquad (9)$$

which can be further simplified into only having one shared PEFT expert, namely **PERFT-S (Single)**

$$\boldsymbol{x}^{1:T} = \sum_{i=1}^{N} \left( G\left(\boldsymbol{h}^{1:T}\right)_i E_i\left(\boldsymbol{h}^{1:T}\right) \right) + \Delta_0\left(\boldsymbol{h}^{1:T}\right) + \boldsymbol{h}^{1:T}, \qquad (10)$$

These two structures implemented the idea of shared experts introduced in recent works (Dai et al., 2024; Qwen, 2024) with PEFT experts, serve as two simpler variants in our PERFT family.

## 4 EXPERIMENTS AND ANALYSES

### 4.1 EXPERIMENT SETUP

**Benchmarks.** Our experiments follow the settings provided by Hu et al. (2023), encompassing 8 benchmarks for commonsense reasoning and 6 for arithmetic reasoning. We utilize their amalgamated training sets Commonsense170K and Math50K to fine-tune models respectively for each domain. Evaluations are conducted correspondingly across all individual benchmark test sets.

**LLM Backbones.** Two state-of-the-art open-source MoE LLMs serve as the backbone models for our experiment: OLMoE-1B-7B (Muennighoff et al., 2024) and Mixtral-8×7B (Jiang et al., 2024). They are selected among publicly available MoE models based on their outstanding performance in the 1B and 13B activated parameter ranges. We use the model weights of their pretrained versions.

**Baselines.** Since there is little previous work on applying PEFT to MoE, we primarily experiment with applying LoRA to attention matrices $\boldsymbol{W}_q$ and $\boldsymbol{W}_v$, the versatile and popular PEFT solution that provides optimal performance under limited parameter budgets (Hu et al., 2022). This serves as our baseline across all scales and tasks. For the smaller OLMoE-1B-7B model, we also include results of applying LoRA to the router matrix $\boldsymbol{W}_g$, as reported in Table 4 in appendix.

**Training.** In our experiments, we maintain consistency with the original training process of each LLM by incorporating their respective auxiliary losses alongside the cross-entropy loss for token

outputs. The models we investigate all include the load balancing loss (Shazeer et al., 2017), which aims to distribute tokens equally among experts. OLMoE-1B-7B additionally incorporates a router z-loss (Zoph et al., 2022b) to penalize large logits in the router for better training stability. To ensure a fair comparison, we keep all auxiliary losses active during fine-tuning for baseline and all PERFT variants. For PERFT-R, we extend this approach with the load balancing loss for the PEFT expert router as well for a similar balanced distribution of tokens among PEFT experts. Detailed hyperparameters and resource configurations for our experiments are provided in Appendix A.

**Design Choices.** For the internal architecture of PERFT and its variants, the major part of our experiments focuses on the application of *parallel LoRA* adapters (He et al., 2022) to the FFN networks, which serves as a simple and effective representation among various possible configurations. The output scaling with $\alpha$ in LoRA also helps us reduce the need to retune hyperparameters when we vary the bottleneck sizes (Yang & Hu, 2020; Hu et al., 2022). For alternative internal architectures, following prior results on dense models (He et al., 2022; Hu et al., 2023), we provide an additional comparative analysis in Appendix B.1 of using vanilla *parallel adapter* (Houlsby et al., 2019; He et al., 2022) with an additional activation function applied between projections.

Regarding routing, we investigate both learned routing (PERFT-R) and embedded routing using the pretrained MoE router (PERFT-E). We also include non-routed variants (PERFT-D and PERFT-S) for comparison. For the number of experts, we explore various configurations as shown in Figure 4. The notation "(TopK/N)" indicates PERFT with $K$ out of $N$ experts activated per forward pass, and "(N)" represents $N$ shared PEFT experts without routing. We examine configurations with the total number of experts ranging from 1 to 64 and activated experts from 1 to 8, allowing us to study the impact of expert count and activation ratio on performance. We experiment with different bottleneck sizes (LoRA ranks) ranging from 2 to 128, as represented by the point sizes in Figure 4. This allows us to study the impact of parameter efficiency on performance across different PERFT variants.

## 4.2 EXPERIMENT RESULTS

Table 1 presents a comparison between several representative PERFT variants and MoE-agnostic baseline with equivalent levels of trainable parameters. The reported PERFT variants consistently outperform baseline methods, with PERFT-R achieving improvements of up to 17.2% and 12.3% on each domain, and PERFT-E up to 10.4% and 5.4%. Section C in appendix provides a comprehensive series of tables detailing the performance of all variants across each individual task.

To obtain the optimal configurations, we conduct an exhaustive series of experiments by fine-tuning OLMoE using combinations of each PERFT variant and possible design choices, with results presented in Figure 4.

| LLM | Arch. | Strategy | # Act. | % Act. | CR | AR |
|---|---|---|---|---|---|---|
| OLMoE 1B-7B (Top8/64) | LoRA$_4$ | $\boldsymbol{W}_q, \boldsymbol{W}_v$@Attn | 0.52M | 0.041 | 57.15 | 28.42 |
| | LoRA$_{16}$ | PERFT-R (Top1/2) | 0.59M | 0.046 | 66.66 | **31.91** |
| | LoRA$_8$ | PERFT-R (Top2/2) | 0.59M | 0.046 | **66.98** | 31.18 |
| | LoRA$_{16}$ | $\boldsymbol{W}_q, \boldsymbol{W}_v$@Attn | 2.10M | 0.164 | 62.86 | 29.71 |
| | LoRA$_4$ | PERFT-E (Top8/64) | 2.10M | 0.164 | **69.42** | 31.30 |
| | LoRA$_{32}$ | PERFT-R (Top1/4) | 2.23M | 0.174 | 67.32 | **32.29** |
| | LoRA$_{64}$ | $\boldsymbol{W}_q, \boldsymbol{W}_v$@Attn | 8.39M | 0.654 | 67.95 | 28.82 |
| | LoRA$_{16}$ | PERFT-E (Top8/64) | 8.39M | 0.654 | **69.29** | 29.08 |
| | LoRA$_{16}$ | PERFT-R (Top8/8) | 8.65M | 0.675 | 68.81 | **31.65** |
| Mixtral 13B-47B (Top2/8) | LoRA$_8$ | $\boldsymbol{W}_q, \boldsymbol{W}_v$@Attn | 3.41M | 0.026 | 85.02 | 64.72 |
| | LoRA$_8$ | PERFT-R (Top2/2) | 4.46M | 0.035 | **86.23** | **69.03** |
| | LoRA$_8$ | PERFT-R (Top2/8) | 5.24M | 0.046 | 85.68 | 68.14 |

Table 1: **Average performance of OLMoE and Mixtral with baseline and PERFT variants on commonsense reasoning (CR) and arithmetic reasoning (AR) benchmarks.** "Arch." denotes the architecture inside PEFT modules. "# Act." and "% Act." represent the number of activated trainable parameters and their ratio to the total activated parameters. "(TopK/N)" refers to activating $K$ experts among the total number of $N$ experts. Performance scores for CR and AR are calculated by averaging the scores across each relevant individual benchmark.

**PERFT-R emerges as the best strategy.** Across both domains, we observe a clear distinction between the overall performance of each PERFT variants. PERFT-R, as expected, emerges as the best strategy that generally outperforms other variants. This advantage is particularly evident at higher levels of parameter efficiency, highlighting its superior potential as an effective strategy for the efficient fine-tuning of MoE models. PERFT-E demonstrates promising performance above the baseline as well. PERFT-S and PERFT-D, as the most simplified variants, fail to yield competitive results across the tested range on both domains.

**PERFT-R and PERFT-E generally benefit from scaling up.** Our results show distinct scaling patterns across different variants of our model. PERFT-R and PERFT-E generally can benefit from scaling up trainable parameters via increased bottleneck sizes $D_B$ within a certain range, as rep-

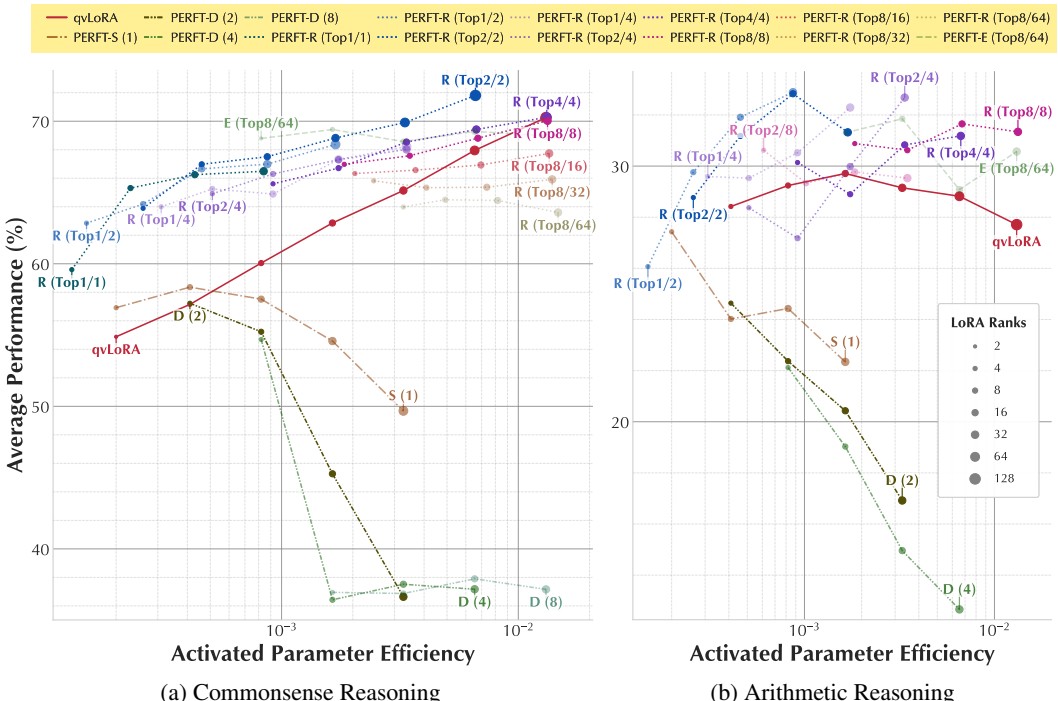

(a) Commonsense Reasoning   (b) Arithmetic Reasoning

Figure 4: **Performance comparison of OLMoE-1B-7B fine-tuned with baselines and PERFT family.** Performance on $y$-axes is averaged across corresponding evaluation benchmarks; "Activated Parameter Efficiency" on $x$-axes indicates the ratio of activated trainable parameters to the total activated parameters. Color represents different methods: "qvLoRA" stands for applying LoRA on attention matrices $W_q$ and $W_v$; "S", "D", "R" and "E" refer to the proposed PERFT variants. Transparency indicates different sparsity levels (ratio of activated experts $K/N$, as "(TopK/N)" labeled for PERFT-R and PERFT-E). Marker size indicates bottleneck size $D_B$.

resented by larger marker sizes in Figure 4. However, PERFT-S and PERFT-D show a rapid performance decline as bottleneck size increases. For the multiplicity of PEFT experts, PERFT-E consistently exhibits performance degradation with more experts, whereas PERFT-R demonstrates a more complex relationship between expert multiplicity and performance, with different trainable parameter ratios yielding varying results.

**PERFT-R is more sensitive to the overall number of PEFT experts.** Figure 5 illustrates the impact of scaling the total number of activated PEFT experts and their trainable parameter efficiencies while controlling for other factors. When fixing the total number of PEFT experts, the performance gain from increasing the activated ratio is relatively modest, suggesting that the performance of PERFT-R is more sensitive to the overall PEFT expert count rather than the proportion activated. It is also observed that on commonsense reasoning tasks, PERFT-R configurations with fewer total PEFT experts tend to outperform those with more experts across various activated parameter efficiencies. In contrast, for math reasoning tasks (Figure 4b), configurations with more PEFT experts do show improved performance as parameter efficiency increases. These divergent patterns reveal that the optimal configuration appears to be task-dependent. Further results on controlling for other factors are provided in Figure 8 in appendix, emphasizing the importance of balancing the total number of experts, sparsity, and computational efficiency when optimizing PERFT configurations for optimal performance.

## 4.3 RESULT ANALYSES

**Routing is important in scaling the number of PEFT experts.** Our experiments reveal fascinating dynamics of PERFT as we manipulate the bottleneck size. As Figure 4 suggests, the optimal information bottleneck configuration represents a delicate balance between capacity and learning effectiveness for each PERFT variant and the given task to achieve peak performance. For PERFT-S and PERFT-D variants without $\tilde{G}(\cdot)$ to distribute gating weights, increasing the bottleneck leads to rapidly decreased average performance across both commonsense and arithmetic reasoning tasks

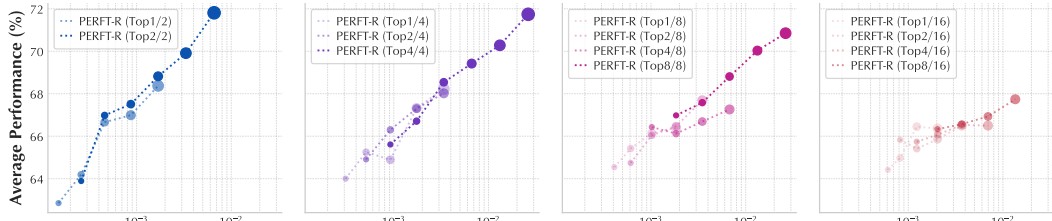

Figure 5: **Performance comparison of configurations with different total number of PEFT experts in PERFT-R.** Results from OLMoE-1B-7B fine-tuned with PERFT-R for commonsense reasoning. $x$-axes stand for activated parameter efficiency. Transparency represents different sparsity levels (ratio of activated PEFT experts), and marker size represents bottleneck size $D_B$.

compared to baseline and other PERFT variants. This phenomenon should be attributed to inefficient parameter utilization in always-activated shared experts. Without an effective routing mechanism, a mismatch would occur between the effective dimensionality of the task and adapter capacity. When the adapter's dimensions significantly exceed the intrinsic dimensionality required by the task for applying modifications, the surplus dimensions in the PEFT module may introduce useless or harmful adaptations, leading to decreased performance as the bottleneck size increases. A detailed discussion on possible reasons is presented in Appendix B.2.

We also observe that naively scaling up the number of experts without a routing mechanism can lead to severe performance degradation. Consistently, PERFT-D underperforms PERFT-S, with performance declining as the number of PERFT experts increases. Figure 6 visualizes this effect through UMAP projections of key memory vectors and expert vectors for the base OLMoE-1B-7B model and different PERFT variants (E, R, D, and S). As the UMAP projection maintains relative distances between original FFN experts in the final results, in an ideal adaptation scenario, PEFT expert key vectors that may activate simultaneously should be distributed evenly within subspaces formed by task-relevant FFN experts' key vectors, maximizing hidden space utilization. However, PERFT-D variants in Figure 6 exhibit tightly clustered key vectors from different experts (shown with different colors), indicating a functional redundancy and inefficient use of model capacity in PERFT-D experts. A detailed analysis on this phenomenon is provided in Appendix B.3.

**Routing contributes more from its weight distribution, rather than sparse activation.** Comparing to PERFT-S and PERFT-D in Figure 4, we observe that even when all experts are activated (Top$N/N$), PERFT-R can still improve the performance significantly, by simply introducing learnable token-wise gating weights for dynamically assigning the importance of each expert's output. This effect is reminiscent of how Gated Linear Units (GLU) improve the FFN layer in transformers (Dauphin et al., 2017). In our case, Figure 6 shows that gating weights can lead to more balanced vector distribution and more effective utilization of hidden space, supporting our discussion in Section 3.1.1. Without such a mechanism, the potential benefits of the increased number of experts may be counterbalanced by the redundancy in model capacity, as discussed in Appendix B.3.

Figure 5 reveals that for a fixed total number of PEFT experts, increasing the sparsity of PERFT-R by activating fewer PEFT experts does not severely degrade performance. This observation is also supported by the visual representation in Figure 6, which suggests that an adequate number of activated expert vectors is sufficient to capture the distribution of the space to be adapted. In addition, the key value vectors from different PEFT experts of PERFT-R that appear clustered in Figure 6 can be utilized by a sparser router to ensure them not activated simultaneously, thereby maintaining performance. This finding indicates that the overall capacity of the PEFT module may be a more critical factor in determining performance rather than the activated capacity.

**With more PEFT experts, PERFT-E can become favored over PERFT-R.** Figure 6 illustrates the distinct dynamics between PERFT-E and PERFT-R. PERFT-E utilizes the frozen expert vectors in the router for FFN experts, while PERFT-R learns an independent router from scratch for PEFT experts. It's important to note that the comparative performance between PERFT-E and PERFT-R can vary in practice, especially when considering scenarios with different activated parameters. Our results in Figure 4a demonstrate that given the same total number of PEFT experts, PERFT-E consistently performs better than PERFT-R (Top8/64) across all bottleneck sizes; while many PERFT-R configurations with fewer experts in turn outperform PERFT-E. When a larger number of PEFT experts are used, utilizing the pretrained router can provide more stable and efficient learning for each

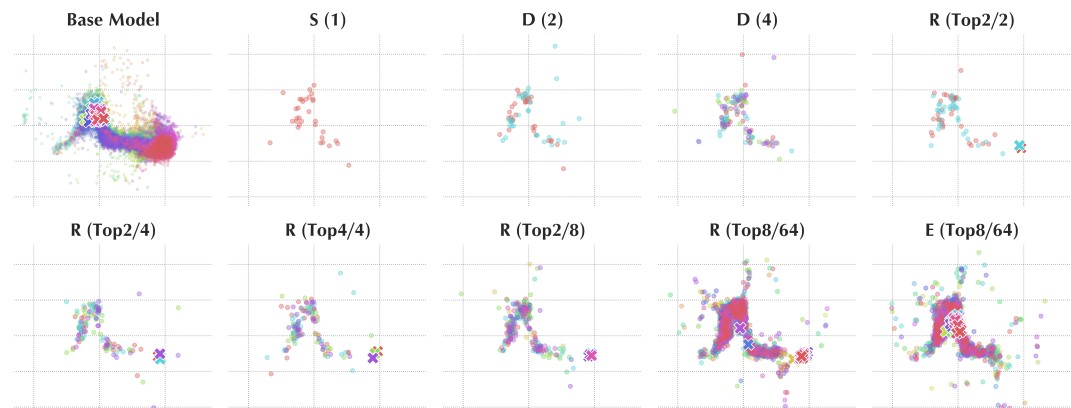

Figure 6: **Visualization of key memory vectors and expert vectors in OLMoE-1B-7B and PERFT family fine-tuned for commonsense reasoning.** Results show projections of vectors with $D_B = 32$ from layer 8 of OLMoE. Each subplot corresponds to a different configuration: "Base Model" showing vectors of FFN experts and router in the original MoE layer; "S", "D", "R" and "E" referring to vectors in the PEFT experts and router (if any) of the corresponding PERFT variants. Markers ● represent key memory vectors in FFN or PEFT experts, and ✖ expert vectors in routers for either FFN experts (in Base Model and PERFT-E) or PEFT experts (in PERFT-R). All vectors are projected using the same PCA and UMAP trained on key memory vectors from the FFN experts. Different colors distinguish vectors associated with different experts.

expert, while PERFT-R may waste more training on exploring larger subspaces and not being able to capture the optimal distribution effectively. This variability highlights the complex trade-off between the flexibility offered by learning new routing mechanisms against the stability gained from utilizing pretrained components in large-scale models, underscoring the need to consider training configuration- and task-specific factors when choosing between them.

## 5 CONCLUSION

In this paper, we introduce a unified framework for integrating PEFT techniques into MoE models, addressing the challenges of efficiently adapting these large, sparse architectures to downstream tasks. Our framework, encompassing both functional and compositional strategies, bridges the gap between existing PEFT methods for dense models and the unique sparsity characteristics of MoE architectures. Building upon this framework, we propose PERFT, a flexible family of PEFT strategies specifically tailored for MoE modules. Through extensive experiments on adapting several state-of-the-art MoE models (OLMoE and Mixtral) for various commonsense and arithmetic reasoning tasks, we demonstrated the effectiveness and scalability of PERFT. Our results showed significant performance improvements over MoE-agnostic baseline methods. We provide an analysis of our findings for each specific design choice from our study, contributing to a deeper understanding of the dynamics between PEFT adaptation strategies and the MoE architecture.

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

## A  TRAINING CONFIGURATIONS

**Hardware.** For each fine-tuning experiment with the baseline and PERFT variant, we trained OLMoE-1B-7B on a single NVIDIA A100 GPU, and Mixtral-8×7B on 4×NVIDIA H100 GPUs using NV-link interconnect across GPUs. Both models are evaluated on NVIDIA A100 GPUs.

**Hyperparameters.** We display the hyperparameter configurations used in fine-tuning and evaluating OLMoE-1B-7B and Mixtral-8×7B in Table 2. We follow Hu et al. (2023) and each model's original settings for training.

## B  ADDITIONAL ANALYSES FOR PERFT CONFIGURATIONS

### B.1  ARCHITECTURE OF PEFT EXPERTS

Table 3 compares the commonsense reasoning performance of LoRA and Parallel Adapters (PA) as PEFT experts in OLMoE-1B-7B with several well-performing PERFT-R configurations. As we

| Hyperparameters | OLMoE-1B-7B | Mixtral-8×7B |
|---|---|---|
| Training precision | BFloat16 | |
| Dropout | 0.05 | |
| Optimizer | AdamW | |
| LR | 1e-5 | 2e-5 |
| LR scheduler | Linear | |
| Batch size | 16 | |
| Warmup steps | 100 | |
| Epochs | 3 | |
| Auxiliary loss coef. | 0.01 | 0.02 |

Table 2: **Hyperparameter configurations for OLMoE-1B-7B and Mixtral-8×7B.**

| Arch. | Strategy | # Act. | % Act. | BoolQ | PIQA | SIQA | HellaS | WinoG | ARC-e | ARC-c | OBQA | Avg. |
|---|---|---|---|---|---|---|---|---|---|---|---|---|
| LoRA$_4$ | PERFT-R (Top1/1) | 0.16M | 0.013 | 62.48 | 75.73 | **68.17** | 25.16 | 51.07 | 76.81 | 55.72 | **61.60** | 59.59 |
| PA$_4$ | PERFT-R (Top1/1) | 0.16M | 0.013 | **63.09** | **76.50** | 64.94 | **31.23** | **52.72** | **77.02** | **56.31** | 55.40 | **59.65** |
| LoRA$_8$ | PERFT-R (Top1/1) | 0.29M | 0.023 | 63.43 | 77.53 | **70.68** | **42.13** | 66.14 | 77.10 | **59.30** | **66.20** | **65.31** |
| PA$_8$ | PERFT-R (Top1/1) | 0.29M | 0.023 | **65.63** | **78.94** | 68.68 | 40.46 | 53.75 | **79.25** | 56.14 | 61.20 | 63.01 |
| LoRA$_{16}$ | PERFT-R (Top1/1) | 0.56M | 0.043 | 64.98 | **78.56** | **72.52** | **41.99** | 67.25 | 77.82 | 58.70 | **68.20** | **66.25** |
| PA$_{16}$ | PERFT-R (Top1/1) | 0.56M | 0.043 | **66.61** | **78.56** | 71.34 | 41.26 | 59.75 | **78.87** | **59.30** | 66.20 | 65.24 |
| LoRA$_{32}$ | PERFT-R (Top1/1) | 1.08M | 0.084 | 66.36 | 78.84 | 72.36 | **42.83** | 63.38 | 78.62 | 58.36 | **71.20** | 66.49 |
| PA$_{32}$ | PERFT-R (Top1/1) | 1.08M | 0.084 | **66.61** | **79.54** | **72.62** | 42.36 | **66.46** | **79.29** | **62.03** | 67.40 | **67.04** |
| LoRA$_4$ | PERFT-R (Top2/2) | 0.33M | 0.026 | 64.86 | 76.71 | **69.60** | 40.89 | **62.43** | 77.23 | 55.80 | **63.60** | **63.89** |
| PA$_4$ | PERFT-R (Top2/2) | 0.33M | 0.026 | **65.44** | **77.48** | 69.40 | **41.14** | 51.54 | **78.83** | **57.94** | 63.20 | 63.12 |
| LoRA$_8$ | PERFT-R (Top2/2) | 0.59M | 0.046 | 65.26 | 78.18 | **72.31** | **42.11** | 71.82 | 77.90 | **60.49** | **67.80** | **66.98** |
| PA$_8$ | PERFT-R (Top2/2) | 0.59M | 0.046 | **67.31** | **80.03** | 71.14 | 41.70 | 61.80 | **78.58** | 58.87 | 66.60 | 65.75 |
| LoRA$_{16}$ | PERFT-R (Top2/2) | 1.11M | 0.087 | 66.18 | 77.97 | **72.52** | **43.99** | 70.64 | 78.24 | 60.75 | 69.80 | 67.51 |
| PA$_{16}$ | PERFT-R (Top2/2) | 1.11M | 0.087 | **66.76** | **79.38** | 72.47 | 43.52 | 69.85 | **80.85** | **61.26** | 71.00 | **68.14** |
| LoRA$_{32}$ | PERFT-R (Top2/2) | 2.16M | 0.169 | 65.81 | 79.38 | **73.59** | **49.42** | **71.59** | 77.78 | **61.18** | 71.80 | 68.82 |
| PA$_{32}$ | PERFT-R (Top2/2) | 2.16M | 0.169 | **67.61** | **80.96** | 73.18 | 45.57 | 70.64 | **80.68** | **61.18** | **72.00** | **68.98** |
| LoRA$_4$ | PERFT-R (Top2/4) | 0.66M | 0.051 | 63.98 | 75.68 | 69.29 | 40.26 | **65.75** | 77.36 | **59.56** | **67.40** | **64.91** |
| PA$_4$ | PERFT-R (Top2/4) | 0.66M | 0.051 | **65.93** | **77.75** | **69.96** | **40.81** | 61.09 | **79.17** | 58.28 | 65.80 | 64.85 |
| LoRA$_8$ | PERFT-R (Top2/4) | 1.18M | 0.092 | **65.02** | 77.86 | **71.90** | 41.61 | 68.75 | 77.31 | 59.13 | **68.80** | 66.30 |
| PA$_8$ | PERFT-R (Top2/4) | 1.18M | 0.092 | 64.40 | **78.07** | 71.24 | **41.80** | 70.17 | **79.76** | **61.09** | 67.80 | **66.79** |
| LoRA$_{16}$ | PERFT-R (Top2/4) | 2.23M | 0.174 | 64.07 | 76.61 | **73.59** | 42.10 | **71.90** | 78.32 | 60.58 | **71.20** | **67.30** |
| PA$_{16}$ | PERFT-R (Top2/4) | 2.23M | 0.174 | **65.99** | **79.92** | 72.62 | **43.14** | 61.64 | **80.09** | 60.58 | 69.20 | 66.65 |
| LoRA$_{32}$ | PERFT-R (Top2/4) | 4.33M | 0.337 | 66.30 | 77.75 | **75.44** | **45.88** | **71.43** | 76.18 | 60.58 | 70.60 | 68.02 |
| PA$_{32}$ | PERFT-R (Top2/4) | 4.33M | 0.337 | **66.70** | **79.33** | 73.18 | 42.57 | 70.40 | **81.10** | **62.20** | 70.60 | **68.26** |

Table 3: **Commonsense reasoning performance of OLMoE-1B-7B with PERFT-R using LoRA and Parallel Adapter (PA) as PEFT experts.** "Arch." denotes the architecture inside PEFT modules. "# Act." and "% Act." represent the number of activated trainable parameters and their ratio to the total activated parameters. "(TopK/N)" refers to activating $K$ experts among the total number of $N$ experts. Dataset names are partially abbreviated, including BoolQ (Clark et al., 2019), PIQA (Bisk et al., 2020), Social IQa (Sap et al., 2019), HellaSwag (Zellers et al., 2019), WinoGrande (Sakaguchi et al., 2021), Easy Set and Challenge Set of ARC (Clark et al., 2018), and OpenBookQA (Mihaylov et al., 2018).

can see, under equivalent activated trainable parameter levels, the average performance difference between LoRA and PA is only marginal. Interestingly, certain architectures consistently outperform others on specific tasks. For instance, parallel adapters generally perform better on BoolQ, PIQA, and ARC, while LoRA excels in SIQA and OBQA. These differences may stem from the inherent nature of knowledge required for each task or specific training data distributions, though a deeper investigation into these task-specific variations is beyond the scope of this study. Given the similar average performance, we opted to focus on LoRA for our experiments due to its simpler structure without the additional activation function.

It is also viable to consider copying the original FFN structure as PEFT experts. We have opted not to investigate this option further in our current study based on two reasons. First, replicating the exact form of FFN experts does not align well with the principles of PEFT, as it would basically become up-scaling the model to a version with more experts. Second, recent advancements have introduced more complex implementations that go beyond the simple $\sigma(hW_{\text{up}})W_{\text{down}}$ pattern how FFN initially

designed as. Gated Linear Unit (GLU), introduced by Dauphin et al. (2017) and Shazeer (2020), has become widely adopted in modern transformers including OLMoE-1B-7B and Mixtral-8×7B. GLU incorporates an additional post-activation gating term $\text{FFN}_{\text{GLU}}(\boldsymbol{h}) = [\sigma(\boldsymbol{h}\boldsymbol{W}_{\text{up}}) \otimes (\boldsymbol{h}\boldsymbol{W}_{\text{gate}})]\boldsymbol{W}_{\text{down}}$, where $\otimes$ denotes element-wise multiplication. The increased complexity of GLU, with its three matrices, presents challenges for a controlled comparison under the same parameter budget. Given these considerations, we focus on experimenting within our current scope.

## B.2 BOTTLENECK SIZE OF PEFT EXPERTS

We provide a detailed empirical analysis about the inefficient parameter utilization when always-activated shared experts are employed without an effective routing mechanism. This symbolizes a mismatch between effective dimensionality and adapter capacity: if the adapter's dimensions significantly exceed the task's intrinsic dimensionality, surplus dimensions may introduce useless or harmful adaptations. Larger random-initialized bottlenecks in PERFT-S and PERFT-D can introduce unnecessary noise in the additional adapted spaces due to insufficient information, interfering with useful representations in the original pretrained model. With the perspective viewing hidden states on the residual stream as bandwidths for modules to communicate on (Elhage et al., 2021), in our PEFT scenario where most parameters remain unchanged, only a relatively small subspace of each layer's hidden state requires task-specific adaptation. Any over-parameterized adaptation can unnecessarily disrupt normal functioning on the residual stream's bandwidths, potentially destabilizing the original gradient flow in the transformer and leading to unstable training or sub-optimal solutions (Aghajanyan et al., 2021). Simultaneously, in the PEFT context with limited adaptation information compared to model pretraining, an excessively large parameter space without gating control can easily result in over-fitting on fine-tuning data, which is exacerbated by the sparse nature of the MoE module we are adapting. As the MoE module hosts multiple different patterns on various combinations of activated FFN experts that dynamically interact with each other on the residual stream, the always-activated PERFT-S and PERFT-D variants may learn unnecessary adaptations during the training process, further aggravating the disrupted functionality and over-fitting problems.

It is also worth noting that since FFN tends to learn task-specific textual patterns (Geva et al., 2021) and attention learns more about positional interactions (Elhage et al., 2021), the nature of different components to which PEFT is introduced also contributes to different phenomena. For the baseline LoRA operating on attention matrices, individual attention heads are already operating on relatively smaller subspaces and can easily write outputs to disjoint subspaces without interaction. The spaces they read and write are relatively more fixed due to the low rank property ($D_{\text{head}} < D$ of hidden space) of multi-head attention matrices. Consequently, additional parameters introduced by scaling the bottleneck of attention LoRA may not interfere with information from other components as severely as adapting the MoE FFN module.

## B.3 MULTIPLICITY OF PEFT EXPERTS WITHOUT ROUTING

This degradation can be explained from the perspective of redundancy in key vector memories. Suppose we have a PERFT-D of $M$ shared experts with bottleneck size $D_B$. This can be viewed as a set of $M$ clusters of key PEFT vectors $\{\tilde{\boldsymbol{e}}_i\}_j, i \in \{1, \cdots, D_B\}, j \in \{1, \cdots, M\}$. At initialization, all weights are randomly distributed. The probability of two randomly chosen vectors being within $\epsilon$ distance of each other can be approximated using the chi-square distribution:

$$p_0(\epsilon) \approx P(\chi^2_{D_B} < \frac{D_B\epsilon^2}{4}) \tag{11}$$

where $\chi^2_{D_B}$ is the chi-square distribution with $D_B$ degrees of freedom. As training progresses, vectors may converge. We can define a factor $\gamma_T$ that represents the increased likelihood of vectors being within $\epsilon$ distance after $T$ training steps:

$$p_T(\epsilon) = \gamma_T \cdot p_0(\epsilon) \tag{12}$$

The expected number of effective vectors after $T$ training steps can be approximated as:

$$E[N_{\text{eff}}(T)] \approx MD_B(1 - e^{-MD_B\gamma_T p_0(\epsilon)^2}) \tag{13}$$

And the efficiency factor:

$$\eta_T(\epsilon) \approx 1 - e^{-MD_B\gamma_T p_0(\epsilon)^2} \tag{14}$$

These formulas depend on $p_0(\epsilon)$, which can be estimated from the initialization distribution, and $\gamma_T$, which represents the cumulative effect of training on vector convergence. The $\gamma_T$ factor encapsulates

the impact of gradient updates over $T$ training steps and could be estimated empirically or through analysis of training dynamics.

## C  ADDITIONAL RESULTS

### C.1  OLMoE-1B-7B FOR COMMONSENSE REASONING

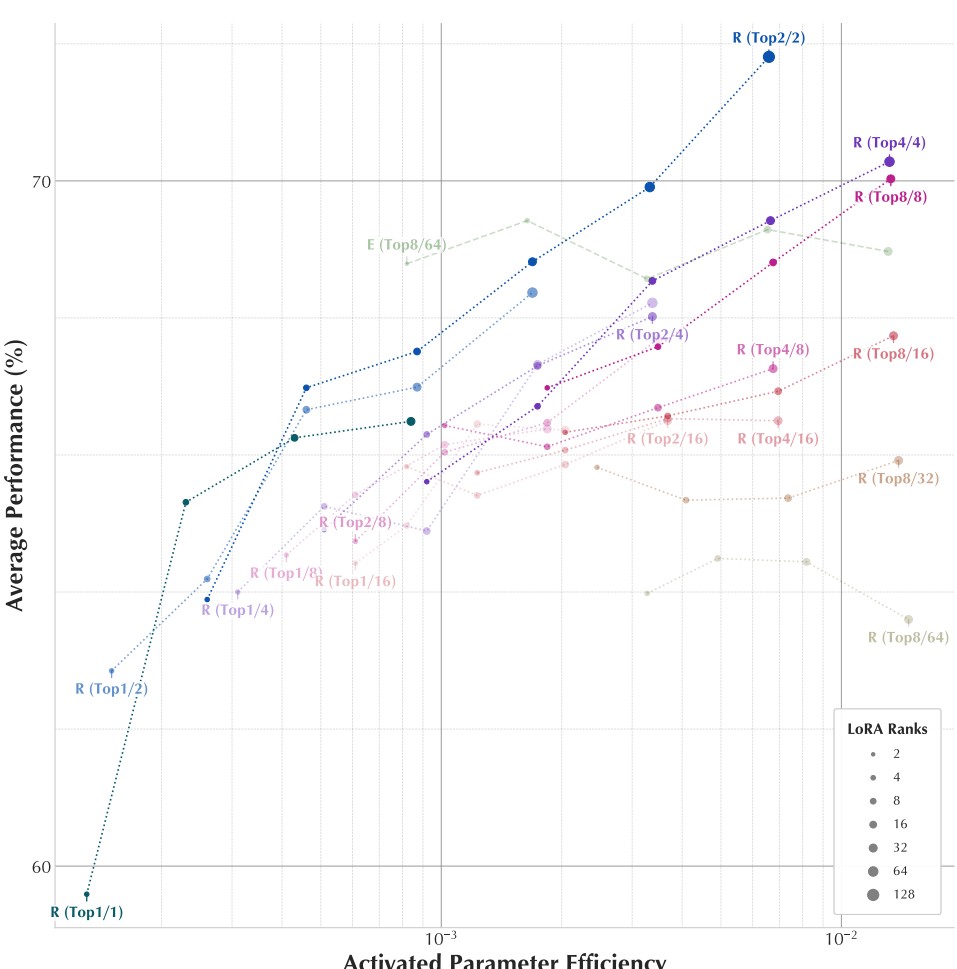

Figure 7: **Performance comparison of OLMoE-1B-7B fine-tuned with different configurations of PERFT-R.** Performance on $y$-axes is averaged across commonsense reasoning benchmarks; "Activated Parameter Efficiency" on $x$-axes indicates the ratio of activated trainable parameters to the total activated parameters. Color represents different configurations of PERFT-R. Transparency indicates different sparsity levels (ratio of activated experts $K/N$, as "(TopK/N)" labeled for PERFT-R and PERFT-E). Marker size indicates bottleneck size $D_B$.

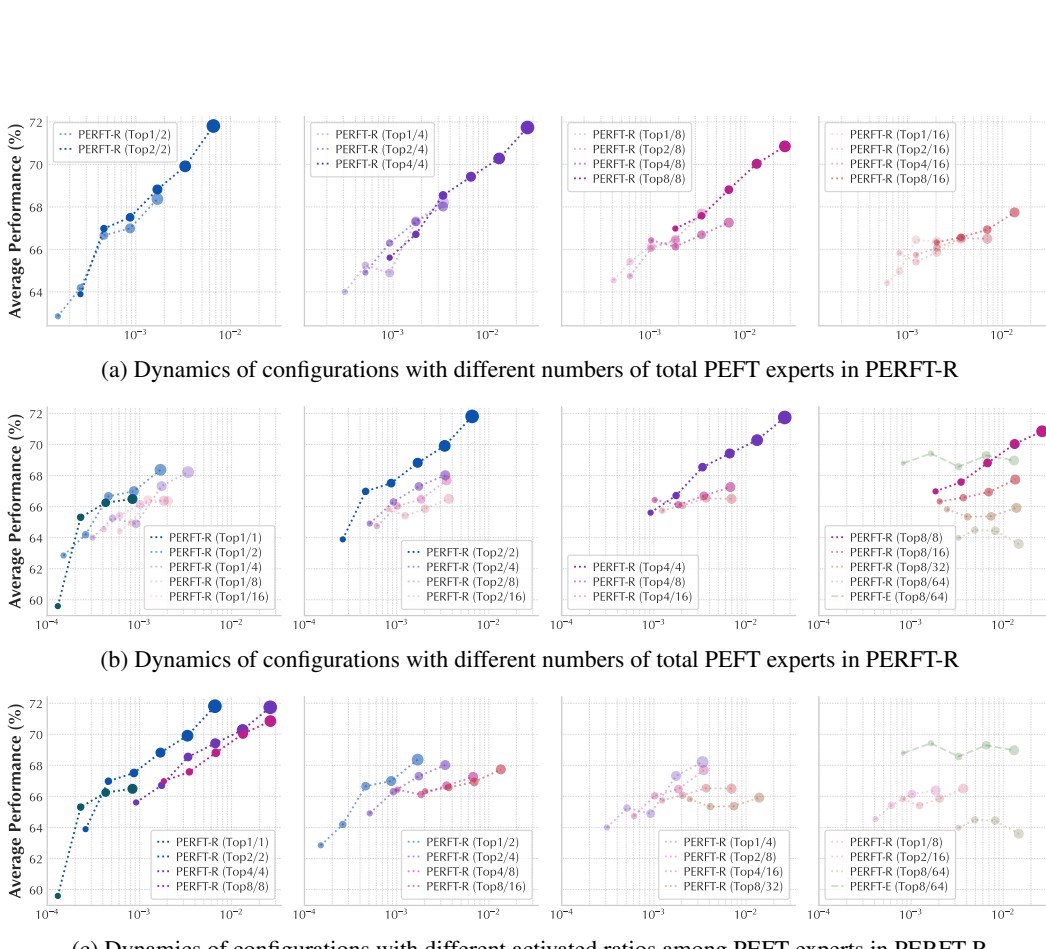

(a) Dynamics of configurations with different numbers of total PEFT experts in PERFT-R

(b) Dynamics of configurations with different numbers of total PEFT experts in PERFT-R

(c) Dynamics of configurations with different activated ratios among PEFT experts in PERFT-R

Figure 8: **Performance comparison of configurations with different total number of PEFT experts in PERFT-R.** Results from OLMoE-1B-7B fine-tuned with PERFT-R for commonsense reasoning. $x$-axes stand for activated parameter efficiency. Transparency represents different sparsity levels (ratio of activated PEFT experts), and marker size represents bottleneck size $D_B$.

| Arch. | Strategy | # Act. | % Act. | BoolQ | PIQA | SIQA | HellaS | WinoG | ARC-e | ARC-c | OBQA | Avg. |
|---|---|---|---|---|---|---|---|---|---|---|---|---|
| Base | (pretrained) | — | — | 42.42 | 52.61 | 16.53 | 21.27 | 28.10 | 13.13 | 13.99 | 6.80 | 24.36 |
| Base | (instruct) | — | — | 59.94 | 62.68 | 12.03 | 22.27 | 5.84 | 15.15 | 17.15 | 8.00 | 25.38 |
| $\text{LoRA}_2$ | $W_q, W_v$@Attn | 0.26M | 0.020 | 62.02 | 71.11 | 59.77 | 28.48 | 50.36 | 70.37 | 48.89 | 48.00 | 54.88 |
| $\text{LoRA}_4$ | $W_q, W_v$@Attn | 0.52M | 0.041 | 60.40 | 73.61 | 62.90 | 32.08 | 50.20 | 74.12 | 52.65 | 51.20 | 57.15 |
| $\text{LoRA}_8$ | $W_q, W_v$@Attn | 1.05M | 0.082 | 63.76 | 74.86 | 65.30 | 37.01 | 50.83 | 76.81 | 55.46 | 56.40 | 60.05 |
| $\text{LoRA}_{16}$ | $W_q, W_v$@Attn | 2.10M | 0.164 | 64.95 | 76.88 | 69.60 | 39.27 | 53.35 | 78.07 | 57.34 | 63.40 | 62.86 |
| $\text{LoRA}_{32}$ | $W_q, W_v$@Attn | 4.19M | 0.327 | 66.79 | 78.56 | 70.93 | 41.63 | 58.41 | 79.38 | 60.41 | 65.00 | 65.14 |
| $\text{LoRA}_{64}$ | $W_q, W_v$@Attn | 8.39M | 0.654 | 67.13 | 80.30 | 73.34 | 44.28 | 65.90 | 80.72 | 61.95 | 70.00 | 67.95 |
| $\text{LoRA}_{128}$ | $W_q, W_v$@Attn | 16.8M | 1.309 | 68.32 | 82.64 | 74.16 | 45.71 | 72.45 | 81.36 | 63.82 | 73.60 | 70.26 |
| $\text{LoRA}_4$ | $W_g$@Gate | 0.14M | 0.011 | 62.14 | 59.79 | 39.66 | 25.94 | 51.62 | 42.63 | 36.52 | 29.00 | 43.41 |
| $\text{LoRA}_8$ | $W_g$@Gate | 0.27M | 0.021 | 59.11 | 66.49 | 47.59 | 27.37 | 51.70 | 52.06 | 42.06 | 33.20 | 47.45 |
| $\text{LoRA}_{16}$ | $W_g$@Gate | 0.54M | 0.042 | 62.05 | 64.04 | 47.85 | 28.08 | 49.33 | 57.37 | 43.17 | 34.40 | 48.29 |
| $\text{LoRA}_{32}$ | $W_g$@Gate | 1.08M | 0.084 | 59.24 | 60.07 | 43.19 | 26.62 | 49.09 | 41.50 | 32.34 | 31.60 | 42.96 |
| $\text{LoRA}_4$ | PERFT-S (1) | 0.26M | 0.020 | 63.82 | 72.31 | 63.87 | 25.45 | 50.12 | 73.91 | 49.49 | 56.40 | 56.92 |
| $\text{LoRA}_8$ | PERFT-S (1) | 0.52M | 0.041 | 63.52 | 73.56 | 66.33 | 25.45 | 51.93 | 72.60 | 52.47 | 61.00 | 58.36 |
| $\text{LoRA}_{16}$ | PERFT-S (1) | 1.05M | 0.082 | 63.49 | 71.71 | 65.71 | 25.11 | 51.22 | 71.13 | 50.60 | 61.20 | 57.52 |
| $\text{LoRA}_{32}$ | PERFT-S (1) | 2.10M | 0.164 | 62.08 | 68.28 | 64.69 | 25.37 | 52.17 | 64.73 | 44.54 | 54.80 | 54.58 |
| $\text{LoRA}_{64}$ | PERFT-S (1) | 4.19M | 0.327 | 61.59 | 63.76 | 59.11 | 24.48 | 54.06 | 53.75 | 36.86 | 43.80 | 49.68 |
| $\text{LoRA}_4$ | PERFT-D (2) | 0.52M | 0.041 | 62.14 | 71.87 | 66.53 | 25.41 | 51.07 | 72.60 | 50.43 | 57.80 | 57.23 |
| $\text{LoRA}_8$ | PERFT-D (2) | 1.05M | 0.082 | 62.87 | 71.44 | 63.41 | 25.47 | 51.70 | 65.28 | 46.84 | 54.80 | 55.23 |
| $\text{LoRA}_{16}$ | PERFT-D (2) | 2.10M | 0.164 | 62.14 | 59.68 | 46.98 | 25.51 | 49.25 | 45.96 | 33.45 | 39.20 | 45.27 |
| $\text{LoRA}_{32}$ | PERFT-D (2) | 4.19M | 0.327 | 62.17 | 48.20 | 32.86 | 25.38 | 48.86 | 24.87 | 25.17 | 25.60 | 36.64 |
| $\text{LoRA}_4$ | PERFT-D (4) | 1.05M | 0.082 | 62.87 | 69.37 | 61.98 | 24.93 | 50.91 | 65.78 | 46.08 | 55.60 | 54.69 |
| $\text{LoRA}_8$ | PERFT-D (4) | 2.10M | 0.164 | 62.17 | 49.29 | 33.06 | 24.57 | 49.57 | 25.46 | 25.09 | 22.20 | 36.43 |
| $\text{LoRA}_{16}$ | PERFT-D (4) | 4.19M | 0.327 | 62.17 | 50.60 | 33.21 | 24.67 | 48.78 | 26.01 | 24.74 | 30.00 | 37.52 |
| $\text{LoRA}_{32}$ | PERFT-D (4) | 8.39M | 0.654 | 62.17 | 52.18 | 33.47 | 25.02 | 50.51 | 25.80 | 22.18 | 26.00 | 37.17 |
| $\text{LoRA}_4$ | PERFT-D (8) | 2.10M | 0.164 | 62.11 | 48.86 | 35.11 | 24.57 | 48.22 | 25.51 | 23.38 | 27.80 | 36.94 |
| $\text{LoRA}_8$ | PERFT-D (8) | 4.19M | 0.327 | 62.17 | 49.13 | 33.27 | 25.37 | 49.41 | 25.00 | 24.23 | 26.40 | 36.87 |
| $\text{LoRA}_{16}$ | PERFT-D (8) | 8.39M | 0.654 | 62.17 | 52.01 | 33.47 | 24.91 | 53.20 | 25.29 | 26.96 | 25.20 | 37.90 |
| $\text{LoRA}_{32}$ | PERFT-D (8) | 16.8M | 1.309 | 62.17 | 50.92 | 33.88 | 24.58 | 49.64 | 24.16 | 26.71 | 25.20 | 37.16 |
| $\text{LoRA}_4$ | PERFT-R (Top1/1) | 0.16M | 0.013 | 62.48 | 75.73 | 68.17 | 25.16 | 51.07 | 76.81 | 55.72 | 61.60 | 59.59 |
| $\text{LoRA}_8$ | PERFT-R (Top1/1) | 0.29M | 0.023 | 63.43 | 77.53 | 70.68 | 42.13 | 66.14 | 77.10 | 59.30 | 66.20 | 65.31 |
| $\text{LoRA}_{16}$ | PERFT-R (Top1/1) | 5.57M | 0.043 | 64.98 | 78.56 | 72.52 | 41.99 | 67.25 | 77.82 | 58.70 | 68.20 | 66.25 |
| $\text{LoRA}_{32}$ | PERFT-R (Top1/1) | 1.08M | 0.084 | 66.36 | 78.84 | 72.36 | 42.83 | 63.38 | 78.62 | 58.36 | 71.20 | 66.49 |
| $\text{LoRA}_4$ | PERFT-R (Top1/2) | 0.20M | 0.015 | 63.67 | 77.04 | 69.09 | 39.92 | 58.09 | 76.81 | 55.80 | 62.40 | 62.85 |
| $\text{LoRA}_8$ | PERFT-R (Top1/2) | 0.33M | 0.026 | 63.98 | 78.13 | 70.93 | 41.00 | 58.88 | 78.11 | 56.66 | 65.80 | 64.19 |
| $\text{LoRA}_{16}$ | PERFT-R (Top1/2) | 0.59M | 0.046 | 65.14 | 76.93 | 72.42 | 41.39 | 70.64 | 78.03 | 59.56 | 69.20 | 66.66 |
| $\text{LoRA}_{32}$ | PERFT-R (Top1/2) | 1.11M | 0.087 | 65.60 | 78.18 | 73.13 | 43.47 | 69.61 | 77.40 | 58.53 | 70.00 | 66.99 |
| $\text{LoRA}_{64}$ | PERFT-R (Top1/2) | 2.16M | 0.169 | 66.09 | 77.97 | 73.75 | 46.36 | 72.61 | 78.79 | 62.20 | 69.20 | 68.37 |
| $\text{LoRA}_4$ | PERFT-R (Top2/2) | 0.33M | 0.026 | 64.86 | 76.71 | 69.60 | 40.89 | 62.43 | 77.23 | 55.80 | 63.60 | 63.89 |
| $\text{LoRA}_8$ | PERFT-R (Top2/2) | 0.59M | 0.046 | 65.26 | 78.18 | 72.31 | 42.11 | 71.82 | 77.90 | 60.49 | 67.80 | 66.99 |
| $\text{LoRA}_{16}$ | PERFT-R (Top2/2) | 1.11M | 0.087 | 66.18 | 77.97 | 72.52 | 43.99 | 70.64 | 78.24 | 60.75 | 69.80 | 67.51 |
| $\text{LoRA}_{32}$ | PERFT-R (Top2/2) | 2.16M | 0.169 | 65.81 | 79.38 | 73.59 | 49.42 | 71.59 | 77.78 | 61.18 | 71.80 | 68.82 |
| $\text{LoRA}_{64}$ | PERFT-R (Top2/2) | 4.26M | 0.332 | 65.96 | 79.87 | 72.82 | 53.93 | 73.40 | 78.91 | 62.20 | 72.20 | 69.91 |
| $\text{LoRA}_{128}$ | PERFT-R (Top2/2) | 8.45M | 0.659 | 67.09 | 80.09 | 74.67 | 68.44 | 70.32 | 79.55 | 60.49 | 73.80 | 71.81 |
| $\text{LoRA}_4$ | PERFT-R (Top1/4) | 0.39M | 0.031 | 63.94 | 76.88 | 69.91 | 39.14 | 60.54 | 78.49 | 57.68 | 65.40 | 64.00 |
| $\text{LoRA}_8$ | PERFT-R (Top1/4) | 0.66M | 0.051 | 64.34 | 77.75 | 71.75 | 40.30 | 67.01 | 77.06 | 58.96 | 64.80 | 65.25 |
| $\text{LoRA}_{16}$ | PERFT-R (Top1/4) | 1.18M | 0.092 | 64.46 | 77.04 | 71.29 | 41.83 | 62.51 | 77.57 | 59.39 | 65.00 | 64.89 |
| $\text{LoRA}_{32}$ | PERFT-R (Top1/4) | 2.23M | 0.174 | 66.21 | 78.51 | 71.49 | 43.87 | 69.61 | 77.69 | 61.01 | 70.20 | 67.32 |
| $\text{LoRA}_{64}$ | PERFT-R (Top1/4) | 4.33 | 0.337 | 65.32 | 79.60 | 73.49 | 45.33 | 71.11 | 77.69 | 62.20 | 71.00 | 68.22 |
| $\text{LoRA}_4$ | PERFT-R (Top2/4) | 0.66M | 0.051 | 63.98 | 75.68 | 69.29 | 40.26 | 65.75 | 77.36 | 59.56 | 67.40 | 64.91 |
| $\text{LoRA}_8$ | PERFT-R (Top2/4) | 1.18M | 0.092 | 65.02 | 77.86 | 71.90 | 41.61 | 68.75 | 77.31 | 59.13 | 68.80 | 66.30 |
| $\text{LoRA}_{16}$ | PERFT-R (Top2/4) | 2.23M | 0.174 | 64.07 | 76.61 | 73.59 | 42.10 | 71.90 | 78.32 | 60.58 | 71.20 | 67.30 |
| $\text{LoRA}_{32}$ | PERFT-R (Top2/4) | 4.33M | 0.337 | 66.30 | 77.75 | 75.44 | 45.88 | 71.43 | 76.18 | 60.58 | 70.60 | 68.02 |
| $\text{LoRA}_4$ | PERFT-R (Top4/4) | 1.18M | 0.092 | 64.25 | 75.84 | 71.03 | 41.40 | 69.22 | 77.65 | 57.08 | 68.40 | 65.61 |
| $\text{LoRA}_8$ | PERFT-R (Top4/4) | 2.23M | 0.174 | 65.14 | 77.64 | 72.98 | 42.67 | 72.45 | 76.98 | 59.39 | 66.40 | 66.71 |
| $\text{LoRA}_{16}$ | PERFT-R (Top4/4) | 4.33M | 0.337 | 65.44 | 79.43 | 73.08 | 48.35 | 71.19 | 77.48 | 59.98 | 73.40 | 68.55 |
| $\text{LoRA}_{32}$ | PERFT-R (Top4/4) | 8.52M | 0.665 | 66.70 | 79.49 | 73.75 | 55.95 | 71.43 | 77.53 | 60.07 | 70.40 | 69.41 |
| $\text{LoRA}_{64}$ | PERFT-R (Top4/4) | 16.9M | 1.319 | 66.02 | 79.71 | 75.49 | 59.29 | 73.32 | 76.64 | 59.90 | 71.80 | 70.27 |
| $\text{LoRA}_{128}$ | PERFT-R (Top4/4) | 33.7M | 2.628 | 65.99 | 78.94 | 75.13 | 67.21 | 73.72 | 78.24 | 59.90 | 74.80 | 71.74 |

Table 4: **(Part 1/2) Evaluation results for OLMoE-1B-7B with baseline methods and PERFT variants on eight commonsense reasoning benchmarks.** "Arch." denotes the architecture inside PEFT modules. "# Act." and "% Act." represent the number of activated trainable parameters and their ratio to the total activated parameters. "(TopK/N)" refers to activating $K$ experts among the total number of $N$ experts. Dataset names are partially abbreviated, including BoolQ (Clark et al., 2019), PIQA (Bisk et al., 2020), Social IQa (Sap et al., 2019), HellaSwag (Zellers et al., 2019), WinoGrande (Sakaguchi et al., 2021), Easy Set and Challenge Set of ARC (Clark et al., 2018), and OpenBookQA (Mihaylov et al., 2018).

| Arch. | Strategy | # Act. | % Act. | BoolQ | PIQA | SIQA | HellaS | WinoG | ARC-e | ARC-c | OBQA | Avg. |
|---|---|---|---|---|---|---|---|---|---|---|---|---|
| LoRA$_4$ | PERFT-R (Top1/8) | 0.52M | 0.041 | 63.73 | 75.30 | 69.91 | 40.77 | 66.77 | 77.69 | 57.51 | 64.60 | 64.54 |
| LoRA$_8$ | PERFT-R (Top1/8) | 0.79M | 0.061 | 64.98 | 77.09 | 70.78 | 41.65 | 66.93 | 77.78 | 57.76 | 66.40 | 65.42 |
| LoRA$_{16}$ | PERFT-R (Top1/8) | 1.31M | 0.102 | 64.89 | 77.26 | 70.88 | 41.95 | 70.09 | 77.31 | 59.39 | 67.40 | 66.15 |
| LoRA$_{32}$ | PERFT-R (Top1/8) | 2.36M | 0.184 | 64.25 | 77.58 | 72.52 | 42.30 | 70.64 | 77.82 | 58.53 | 67.40 | 66.38 |
| LoRA$_4$ | PERFT-R (Top2/8) | 0.79M | 0.061 | 64.28 | 76.99 | 68.88 | 40.61 | 66.85 | 77.57 | 57.34 | 65.40 | 64.74 |
| LoRA$_8$ | PERFT-R (Top2/8) | 1.31M | 0.102 | 63.91 | 76.88 | 71.03 | 43.45 | 69.69 | 77.23 | 58.11 | 68.00 | 66.04 |
| LoRA$_{16}$ | PERFT-R (Top2/8) | 2.36M | 0.184 | 64.68 | 77.64 | 72.36 | 43.33 | 71.51 | 75.97 | 58.45 | 67.80 | 66.47 |
| LoRA$_{32}$ | PERFT-R (Top2/8) | 4.46M | 0.348 | 64.40 | 78.13 | 74.21 | 46.80 | 71.59 | 76.39 | 58.79 | 71.20 | 67.69 |
| LoRA$_4$ | PERFT-R (Top4/8) | 1.31M | 0.102 | 64.74 | 77.04 | 71.60 | 42.82 | 70.01 | 77.31 | 59.73 | 68.20 | 66.43 |
| LoRA$_8$ | PERFT-R (Top4/8) | 2.36M | 0.184 | 64.86 | 76.61 | 73.69 | 42.10 | 69.46 | 76.98 | 58.02 | 67.20 | 66.12 |
| LoRA$_{16}$ | PERFT-R (Top4/8) | 4.46M | 0.348 | 65.78 | 76.33 | 72.57 | 45.61 | 69.53 | 76.22 | 58.28 | 69.20 | 66.69 |
| LoRA$_{32}$ | PERFT-R (Top4/8) | 8.65M | 0.675 | 65.20 | 77.37 | 73.64 | 46.36 | 72.45 | 77.02 | 56.83 | 69.20 | 67.26 |
| LoRA$_4$ | PERFT-R (Top8/8) | 2.36M | 0.184 | 64.98 | 77.37 | 72.77 | 45.71 | 70.32 | 77.15 | 58.96 | 68.60 | 66.98 |
| LoRA$_8$ | PERFT-R (Top8/8) | 4.46M | 0.348 | 64.98 | 78.13 | 74.21 | 46.75 | 69.85 | 77.19 | 59.56 | 70.00 | 67.58 |
| LoRA$_{16}$ | PERFT-R (Top8/8) | 8.65M | 0.675 | 65.93 | 77.58 | 74.41 | 55.14 | 71.98 | 76.47 | 57.59 | 71.40 | 68.81 |
| LoRA$_{32}$ | PERFT-R (Top8/8) | 17.0M | 1.329 | 65.78 | 78.07 | 74.92 | 58.44 | 71.82 | 76.05 | 61.35 | 73.80 | 70.03 |
| LoRA$_{64}$ | PERFT-R (Top8/8) | 33.8M | 2.638 | 65.20 | 80.25 | 75.13 | 65.68 | 73.01 | 75.67 | 59.47 | 72.40 | 70.85 |
| LoRA$_4$ | PERFT-R (Top1/16) | 0.79M | 0.061 | 64.65 | 75.73 | 70.83 | 40.04 | 63.61 | 77.06 | 59.04 | 64.40 | 64.42 |
| LoRA$_8$ | PERFT-R (Top1/16) | 1.05M | 0.082 | 64.98 | 76.17 | 69.60 | 40.17 | 67.48 | 76.30 | 58.02 | 67.00 | 64.97 |
| LoRA$_{16}$ | PERFT-R (Top1/16) | 1.57M | 0.123 | 63.79 | 77.04 | 73.29 | 42.39 | 70.56 | 76.60 | 58.96 | 69.00 | 66.45 |
| LoRA$_{32}$ | PERFT-R (Top1/16) | 2.62M | 0.204 | 64.25 | 75.79 | 72.21 | 43.98 | 70.24 | 76.18 | 59.04 | 69.20 | 66.36 |
| LoRA$_4$ | PERFT-R (Top2/16) | 1.05M | 0.082 | 63.94 | 77.31 | 71.44 | 41.23 | 69.22 | 78.37 | 58.11 | 67.00 | 65.83 |
| LoRA$_8$ | PERFT-R (Top2/16) | 1.57M | 0.123 | 62.45 | 76.12 | 71.55 | 41.75 | 67.80 | 76.14 | 59.47 | 68.00 | 65.41 |
| LoRA$_{16}$ | PERFT-R (Top2/16) | 2.62M | 0.204 | 64.50 | 76.06 | 71.03 | 43.21 | 69.22 | 75.59 | 59.30 | 68.00 | 65.86 |
| LoRA$_{32}$ | PERFT-R (Top2/16) | 4.72M | 0.368 | 65.35 | 76.50 | 72.98 | 47.08 | 69.30 | 74.79 | 58.19 | 67.80 | 66.50 |
| LoRA$_4$ | PERFT-R (Top4/16) | 1.57M | 0.123 | 64.37 | 75.52 | 72.36 | 42.12 | 69.61 | 76.35 | 57.59 | 68.00 | 65.74 |
| LoRA$_8$ | PERFT-R (Top4/16) | 2.62M | 0.204 | 64.92 | 76.55 | 72.21 | 43.09 | 69.61 | 75.67 | 59.30 | 67.20 | 66.07 |
| LoRA$_{16}$ | PERFT-R (Top4/16) | 4.72M | 0.368 | 65.50 | 76.50 | 73.80 | 43.82 | 71.43 | 74.03 | 57.34 | 69.80 | 66.53 |
| LoRA$_{32}$ | PERFT-R (Top4/16) | 8.91M | 0.695 | 65.47 | 77.09 | 73.64 | 45.04 | 69.77 | 74.49 | 58.70 | 67.80 | 66.50 |
| LoRA$_4$ | PERFT-R (Top8/16) | 2.62M | 0.204 | 64.25 | 76.06 | 72.31 | 41.46 | 71.11 | 76.81 | 60.67 | 68.00 | 66.33 |
| LoRA$_8$ | PERFT-R (Top8/16) | 4.72M | 0.368 | 64.50 | 77.53 | 73.34 | 45.22 | 71.74 | 74.92 | 57.51 | 67.80 | 66.57 |
| LoRA$_{16}$ | PERFT-R (Top8/16) | 8.91M | 0.695 | 64.53 | 77.91 | 73.54 | 47.24 | 71.27 | 75.00 | 54.78 | 71.20 | 66.93 |
| LoRA$_{32}$ | PERFT-R (Top8/16) | 17.3M | 1.350 | 65.57 | 76.82 | 74.51 | 53.13 | 70.01 | 74.07 | 57.17 | 70.60 | 67.73 |
| LoRA$_4$ | PERFT-R (Top8/32) | 3.15M | 0.245 | 63.82 | 75.52 | 72.57 | 41.75 | 72.30 | 74.37 | 57.25 | 69.00 | 65.82 |
| LoRA$_8$ | PERFT-R (Top8/32) | 5.24M | 0.409 | 63.79 | 75.35 | 71.70 | 43.90 | 67.88 | 74.03 | 58.28 | 67.80 | 65.34 |
| LoRA$_{16}$ | PERFT-R (Top8/32) | 9.44M | 0.736 | 64.07 | 75.90 | 73.39 | 44.59 | 72.22 | 72.31 | 55.29 | 65.20 | 65.37 |
| LoRA$_{32}$ | PERFT-R (Top8/32) | 17.8M | 1.390 | 64.71 | 75.35 | 73.95 | 47.17 | 70.72 | 72.22 | 55.46 | 67.80 | 65.92 |
| LoRA$_4$ | PERFT-R (Top8/64) | 4.19M | 0.327 | 63.55 | 76.06 | 70.11 | 42.16 | 69.14 | 72.31 | 53.67 | 64.80 | 63.98 |
| LoRA$_8$ | PERFT-R (Top8/64) | 6.29M | 0.491 | 64.53 | 75.52 | 72.21 | 41.79 | 70.40 | 71.38 | 53.92 | 66.20 | 64.49 |
| LoRA$_{16}$ | PERFT-R (Top8/64) | 10.5M | 0.818 | 64.71 | 73.61 | 72.26 | 42.35 | 70.88 | 71.09 | 54.78 | 64.80 | 64.44 |
| LoRA$_{32}$ | PERFT-R (Top8/64) | 18.9M | 1.472 | 62.81 | 74.43 | 72.31 | 41.11 | 69.22 | 69.49 | 53.84 | 65.60 | 63.60 |
| LoRA$_2$ | PERFT-E (Top8/64) | 1.05M | 0.082 | 65.54 | 79.11 | 73.59 | 50.06 | 73.24 | 77.27 | 58.70 | 72.80 | 68.79 |
| LoRA$_4$ | PERFT-E (Top8/64) | 2.10M | 0.164 | 64.80 | 79.49 | 74.36 | 58.39 | 72.69 | 75.00 | 58.45 | 72.20 | 69.42 |
| LoRA$_8$ | PERFT-E (Top8/64) | 4.19M | 0.327 | 65.81 | 78.84 | 73.85 | 58.84 | 71.51 | 74.41 | 56.06 | 69.20 | 68.56 |
| LoRA$_{16}$ | PERFT-E (Top8/64) | 8.39M | 0.654 | 65.20 | 78.24 | 74.97 | 64.35 | 72.30 | 74.41 | 55.46 | 69.40 | 69.29 |
| LoRA$_{32}$ | PERFT-E (Top8/64) | 16.8M | 1.309 | 66.51 | 76.39 | 74.26 | 62.55 | 73.09 | 72.22 | 56.14 | 70.60 | 68.97 |
| LoRA$_{64}$ | PERFT-E (Top8/64) | 33.6M | 2.617 | 65.57 | 77.09 | 73.80 | 59.89 | 73.32 | 71.72 | 56.40 | 68.80 | 68.32 |

Table 5: **(Part 2/2) Evaluation results for OLMoE-1B-7B with baseline methods and PERFT variants on eight commonsense reasoning benchmarks.** "Arch." denotes the architecture inside PEFT modules. "# Act." and "% Act." represent the number of activated trainable parameters and their ratio to the total activated parameters. "(TopK/N)" refers to activating $K$ experts among the total number of $N$ experts. Dataset names are partially abbreviated, including BoolQ (Clark et al., 2019), PIQA (Bisk et al., 2020), Social IQa (Sap et al., 2019), HellaSwag (Zellers et al., 2019), WinoGrande (Sakaguchi et al., 2021), Easy Set and Challenge Set of ARC (Clark et al., 2018), and OpenBookQA (Mihaylov et al., 2018).

## C.2 OLMoE-1B-7B FOR ARITHMETIC REASONING

| Arch. | Strategy | # Act. | % Act. | MultiArith | GSM8K | AddSub | AQuA | SingleEq | SVAMP | Avg. |
|---|---|---|---|---|---|---|---|---|---|---|
| $\text{LoRA}_2$ | $\boldsymbol{W}_q, \boldsymbol{W}_v$@Attn | 0.26M | 0.020 | 20.00 | 8.72 | 43.04 | 20.47 | 52.95 | 29.40 | 29.10 |
| $\text{LoRA}_4$ | $\boldsymbol{W}_q, \boldsymbol{W}_v$@Attn | 0.52M | 0.041 | 21.83 | 8.11 | 40.51 | 20.47 | 50.79 | 28.80 | 28.42 |
| $\text{LoRA}_8$ | $\boldsymbol{W}_q, \boldsymbol{W}_v$@Attn | 1.05M | 0.082 | 17.33 | 8.57 | 44.05 | 24.02 | 50.59 | 30.90 | 29.24 |
| $\text{LoRA}_{16}$ | $\boldsymbol{W}_q, \boldsymbol{W}_v$@Attn | 2.10M | 0.164 | 18.83 | 9.02 | 46.58 | 24.02 | 50.59 | 29.20 | 29.71 |
| $\text{LoRA}_{32}$ | $\boldsymbol{W}_q, \boldsymbol{W}_v$@Attn | 4.19M | 0.327 | 19.17 | 8.79 | 43.54 | 23.23 | 51.97 | 28.20 | 29.15 |
| $\text{LoRA}_{64}$ | $\boldsymbol{W}_q, \boldsymbol{W}_v$@Attn | 8.39M | 0.654 | 17.00 | 9.10 | 47.09 | 22.83 | 49.80 | 27.10 | 28.82 |
| $\text{LoRA}_{128}$ | $\boldsymbol{W}_q, \boldsymbol{W}_v$@Attn | 16.8M | 1.309 | 15.00 | 8.11 | 44.81 | 22.83 | 49.02 | 26.50 | 27.71 |
| $\text{LoRA}_4$ | PERFT-S (1) | 0.26M | 0.020 | 21.00 | 5.61 | 40.00 | 18.50 | 50.59 | 28.90 | 27.43 |
| $\text{LoRA}_8$ | PERFT-S (1) | 0.52M | 0.041 | 17.00 | 6.22 | 34.18 | 17.32 | 30.20 | 27.30 | 24.02 |
| $\text{LoRA}_{16}$ | PERFT-S (1) | 1.05M | 0.082 | 14.83 | 6.29 | 35.19 | 21.26 | 41.73 | 27.30 | 24.43 |
| $\text{LoRA}_{32}$ | PERFT-S (1) | 2.10M | 0.164 | 16.17 | 4.09 | 34.68 | 18.11 | 37.40 | 23.60 | 22.34 |
| $\text{LoRA}_4$ | PERFT-D (2) | 0.52M | 0.041 | 18.67 | 5.76 | 37.97 | 20.08 | 40.75 | 24.60 | 24.64 |
| $\text{LoRA}_8$ | PERFT-D (2) | 1.05M | 0.082 | 15.67 | 5.46 | 33.16 | 18.11 | 37.40 | 24.40 | 22.37 |
| $\text{LoRA}_{16}$ | PERFT-D (2) | 2.10M | 0.164 | 14.00 | 4.85 | 30.13 | 16.93 | 34.65 | 22.00 | 20.43 |
| $\text{LoRA}_{32}$ | PERFT-D (2) | 4.19M | 0.327 | 8.17 | 3.87 | 29.11 | 19.29 | 25.39 | 15.70 | 16.92 |
| $\text{LoRA}_4$ | PERFT-D (4) | 1.05M | 0.082 | 14.17 | 5.08 | 34.18 | 22.05 | 35.43 | 21.80 | 22.12 |
| $\text{LoRA}_8$ | PERFT-D (4) | 2.10M | 0.164 | 9.17 | 3.94 | 31.65 | 19.69 | 29.13 | 20.60 | 19.03 |
| $\text{LoRA}_{16}$ | PERFT-D (4) | 4.19M | 0.327 | 9.33 | 3.03 | 21.77 | 20.87 | 21.46 | 13.30 | 14.96 |
| $\text{LoRA}_{32}$ | PERFT-D (4) | 8.39M | 0.654 | 4.33 | 1.97 | 16.20 | 21.65 | 18.90 | 12.90 | 12.66 |
| $\text{LoRA}_4$ | PERFT-R (Top1/2) | 0.20M | 0.015 | 18.83 | 7.88 | 41.77 | 16.93 | 44.88 | 26.10 | 26.07 |
| $\text{LoRA}_8$ | PERFT-R (Top1/2) | 0.33M | 0.026 | 19.00 | 7.51 | 47.09 | 19.69 | 53.35 | 31.90 | 29.75 |
| $\text{LoRA}_{16}$ | PERFT-R (Top1/2) | 0.59M | 0.046 | 21.17 | 8.79 | 52.15 | 19.69 | 57.68 | 32.00 | 31.91 |
| $\text{LoRA}_{32}$ | PERFT-R (Top1/2) | 1.11M | 0.087 | 27.17 | 9.33 | 50.89 | 20.87 | 57.09 | 32.00 | 32.89 |
| $\text{LoRA}_4$ | PERFT-R (Top2/2) | 0.33M | 0.026 | 21.17 | 8.19 | 45.82 | 18.11 | 49.02 | 30.30 | 28.77 |
| $\text{LoRA}_8$ | PERFT-R (Top2/2) | 0.59M | 0.046 | 23.33 | 7.35 | 51.65 | 18.50 | 52.76 | 33.50 | 31.18 |
| $\text{LoRA}_{16}$ | PERFT-R (Top2/2) | 1.11M | 0.087 | 26.50 | 8.49 | 52.15 | 20.87 | 56.69 | 32.30 | 32.83 |
| $\text{LoRA}_{32}$ | PERFT-R (Top2/2) | 2.16M | 0.169 | 23.67 | 9.25 | 44.81 | 21.65 | 53.35 | 35.20 | 31.32 |
| $\text{LoRA}_4$ | PERFT-R (Top1/4) | 0.39M | 0.031 | 18.83 | 8.87 | 48.86 | 21.65 | 50.20 | 29.10 | 29.59 |
| $\text{LoRA}_8$ | PERFT-R (Top1/4) | 0.66M | 0.051 | 20.83 | 9.48 | 44.05 | 17.32 | 55.91 | 29.60 | 29.53 |
| $\text{LoRA}_{16}$ | PERFT-R (Top1/4) | 1.18M | 0.092 | 22.67 | 7.88 | 46.84 | 20.47 | 51.77 | 33.50 | 30.52 |
| $\text{LoRA}_{32}$ | PERFT-R (Top1/4) | 2.23M | 0.174 | 25.67 | 7.35 | 54.18 | 19.69 | 54.72 | 32.10 | 32.28 |
| $\text{LoRA}_4$ | PERFT-R (Top2/4) | 0.66M | 0.051 | 19.33 | 7.73 | 45.32 | 16.93 | 49.21 | 31.70 | 28.37 |
| $\text{LoRA}_8$ | PERFT-R (Top2/4) | 1.18M | 0.092 | 16.33 | 6.97 | 44.30 | 16.54 | 48.82 | 30.10 | 27.18 |
| $\text{LoRA}_{16}$ | PERFT-R (Top2/4) | 2.23M | 0.174 | 20.83 | 8.34 | 47.34 | 18.50 | 51.18 | 33.70 | 29.98 |
| $\text{LoRA}_{32}$ | PERFT-R (Top2/4) | 4.33M | 0.337 | 28.00 | 9.10 | 49.37 | 19.29 | 57.09 | 33.20 | 32.67 |
| $\text{LoRA}_4$ | PERFT-R (Top4/4) | 1.18M | 0.092 | 20.67 | 7.58 | 47.85 | 20.08 | 53.35 | 31.30 | 30.14 |
| $\text{LoRA}_8$ | PERFT-R (Top4/4) | 2.23M | 0.174 | 25.33 | 7.73 | 40.51 | 20.08 | 49.02 | 30.70 | 28.89 |
| $\text{LoRA}_{16}$ | PERFT-R (Top4/4) | 4.33M | 0.337 | 21.50 | 7.43 | 45.06 | 20.87 | 59.84 | 30.30 | 30.83 |
| $\text{LoRA}_{32}$ | PERFT-R (Top4/4) | 8.52M | 0.665 | 22.17 | 8.34 | 50.38 | 20.08 | 55.31 | 30.80 | 31.18 |
| $\text{LoRA}_4$ | PERFT-R (Top2/8) | 0.79M | 0.061 | 21.83 | 7.88 | 50.89 | 21.26 | 51.97 | 29.90 | 30.62 |
| $\text{LoRA}_8$ | PERFT-R (Top2/8) | 1.31M | 0.102 | 20.00 | 8.26 | 47.34 | 19.29 | 52.76 | 28.30 | 29.33 |
| $\text{LoRA}_{16}$ | PERFT-R (Top2/8) | 2.36M | 0.184 | 22.33 | 8.72 | 46.08 | 20.87 | 50.39 | 30.20 | 29.76 |
| $\text{LoRA}_{32}$ | PERFT-R (Top2/8) | 4.46M | 0.348 | 22.50 | 7.43 | 46.84 | 18.90 | 50.59 | 30.90 | 29.53 |
| $\text{LoRA}_4$ | PERFT-R (Top8/8) | 2.36M | 0.184 | 28.33 | 7.81 | 47.85 | 16.93 | 53.15 | 31.20 | 30.88 |
| $\text{LoRA}_8$ | PERFT-R (Top8/8) | 4.46M | 0.348 | 21.00 | 8.49 | 49.37 | 21.26 | 51.97 | 31.60 | 30.61 |
| $\text{LoRA}_{16}$ | PERFT-R (Top8/8) | 8.65M | 0.675 | 28.50 | 8.04 | 45.82 | 20.87 | 53.74 | 32.90 | 31.64 |
| $\text{LoRA}_{32}$ | PERFT-R (Top8/8) | 17.0M | 1.329 | 27.67 | 8.49 | 45.06 | 21.26 | 52.95 | 32.60 | 31.34 |
| $\text{LoRA}_4$ | PERFT-E (Top8/64) | 2.10M | 0.164 | 26.67 | 6.44 | 46.58 | 22.05 | 53.94 | 32.10 | 31.30 |
| $\text{LoRA}_8$ | PERFT-E (Top8/64) | 4.19M | 0.327 | 28.33 | 7.81 | 43.80 | 21.26 | 57.28 | 32.60 | 31.85 |
| $\text{LoRA}_{16}$ | PERFT-E (Top8/64) | 8.39M | 0.654 | 25.17 | 8.42 | 43.29 | 19.29 | 48.82 | 29.50 | 29.08 |
| $\text{LoRA}_{32}$ | PERFT-E (Top8/64) | 16.8M | 1.309 | 26.17 | 6.75 | 44.05 | 20.87 | 52.76 | 32.80 | 30.56 |

Table 6: **Evaluation results for OLMoE-1B-7B with baseline methods and PERFT variants on six arithmetic reasoning benchmarks.** "Arch." denotes the architecture inside PEFT modules. "# Act." and "% Act." represent the number of activated trainable parameters and their ratio to the total activated parameters. "(TopK/N)" refers to activating $K$ experts among the total number of $N$ experts. Dataset names are partially abbreviated, including MultiArith (Roy & Roth, 2015), GSM8K (Cobbe et al., 2021), AddSub (Hosseini et al., 2014), AQuA (Ling et al., 2017), SingleEq (Koncel-Kedziorski et al., 2015), and SVAMP (Patel et al., 2021).

## C.3 Mixtral-8×7B for Commonsense Reasoning

| Arch. | Strategy | # Act. | % Act. | BoolQ | PIQA | SIQA | HellaS | WinoG | ARC-e | ARC-c | OBQA | Avg. |
|---|---|---|---|---|---|---|---|---|---|---|---|---|
| Base | (pretrained) | — | — | 51.10 | 81.12 | 46.11 | 47.54 | 49.88 | 53.20 | 52.99 | 39.20 | 52.64 |
| Base | (instruct) | — | — | 68.87 | 88.30 | 68.58 | 72.06 | 59.98 | 89.52 | 78.50 | 74.40 | 75.03 |
| LoRA$_8$ | $\boldsymbol{W}_q, \boldsymbol{W}_v$@Attn | 3.41M | 0.026 | 73.49 | 90.04 | 81.17 | 89.67 | 82.16 | 93.56 | 83.87 | 86.20 | 85.02 |
| LoRA$_{16}$ | PERFT-S (1) | 4.19M | 0.033 | 75.11 | 90.26 | 81.63 | 94.26 | 84.85 | 92.85 | 81.40 | 87.60 | 85.99 |
| LoRA$_8$ | PERFT-R (Top2/2) | 4.46M | 0.035 | 74.68 | 89.77 | 81.47 | 94.33 | 86.27 | 92.05 | 81.48 | 89.80 | 86.23 |
| LoRA$_{16}$ | PERFT-R (Top1/4) | 4.72M | 0.037 | 72.84 | 89.12 | 80.40 | 92.69 | 84.37 | 91.84 | 82.25 | 85.80 | 84.91 |
| LoRA$_8$ | PERFT-R (Top2/4) | 4.72M | 0.037 | 74.71 | 90.10 | 79.38 | 94.18 | 85.71 | 92.09 | 81.31 | 85.80 | 85.41 |
| LoRA$_8$ | PERFT-R (Top2/8) | 5.24M | 0.041 | 73.76 | 89.12 | 81.63 | 94.51 | 85.16 | 91.67 | 80.20 | 87.80 | 85.48 |
| LoRA$_8$ | PERFT-E (Top2/8) | 4.19M | 0.033 | 74.13 | 90.21 | 80.81 | 91.36 | 86.42 | 92.21 | 81.06 | 88.60 | 85.60 |

Table 7: **Evaluation results for Mixtral-8×7B with baseline methods and PERFT variants on eight commonsense reasoning benchmarks.** "Arch." denotes the architecture inside PEFT modules. "# Act." and "% Act." represent the number of activated trainable parameters and their ratio to the total activated parameters. "(TopK/N)" refers to activating $K$ experts among the total number of $N$ experts. Dataset names are partially abbreviated, including BoolQ (Clark et al., 2019), PIQA (Bisk et al., 2020), Social IQa (Sap et al., 2019), HellaSwag (Zellers et al., 2019), WinoGrande (Sakaguchi et al., 2021), Easy Set and Challenge Set of ARC (Clark et al., 2018), and OpenBookQA (Mihaylov et al., 2018).

## C.4 Mixtral-8×7B for Arithmetic Reasoning

| Arch. | Strategy | # Act. | % Act. | MultiArith | GSM8K | AddSub | AQuA | SingleEq | SVAMP | Avg. |
|---|---|---|---|---|---|---|---|---|---|---|
| LoRA$_8$ | $\boldsymbol{W}_q, \boldsymbol{W}_v$@Attn | 3.41M | 0.026 | 60.00 | 50.87 | 90.13 | 28.74 | 89.37 | 69.20 | 64.72 |
| LoRA$_8$ | PERFT-R (Top2/2) | 4.46M | 0.035 | 82.83 | 55.80 | 87.59 | 29.92 | 89.76 | 68.30 | 69.04 |
| LoRA$_8$ | PERFT-R (Top2/8) | 5.24M | 0.041 | 79.00 | 54.06 | 87.34 | 29.13 | 88.98 | 70.30 | 68.13 |

Table 8: **Evaluation results for Mixtral-8×7B with baseline methods and PERFT variants on six arithmetic reasoning benchmarks.** "Arch." denotes the architecture inside PEFT modules. "# Act." and "% Act." represent the number of activated trainable parameters and their ratio to the total activated parameters. "(TopK/N)" refers to activating $K$ experts among the total number of $N$ experts. Dataset names are partially abbreviated, including MultiArith (Roy & Roth, 2015), GSM8K (Cobbe et al., 2021), AddSub (Hosseini et al., 2014), AQuA (Ling et al., 2017), SingleEq (Koncel-Kedziorski et al., 2015), and SVAMP (Patel et al., 2021).

