# OpenReview forum: "PERFT: Parameter-Efficient Routed Fine-Tuning for Mixture-of-Expert Model"
_ICLR.cc/2025/Conference — Submitted to ICLR 2025_

### Official Review · Reviewer_SUvZ · 2024-10-24

**Soundness:** 3
**Presentation:** 3
**Contribution:** 3
**Rating:** 8
**Confidence:** 4

**Summary:**

The paper introduces a PEFT framework tailored for Mixture-of-Experts (MoE) models.  Within the realm of functional strategies, the paper delves into the internal workings of PEFT modules, encompassing the architecture of individual PEFT experts, the multiplicity of these experts, and the routing mechanisms among them.  On the other hand, the compositional strategies dimension elucidates how PEFT modules interact with the original MoE architecture, exploring configurations such as shared or embedded PEFT experts. The empirical validation through diverse experiments showcases substantial performance enhancements across tasks like commonsense reasoning and arithmetic word problems, underscoring the framework's efficacy in enhancing model training efficiency and scalability.

**Strengths:**

1. This paper makes some efforts to design some strategies for MoE's PEFT. It  can serve as a benchmark or an empirical report to the community.

2. The authors detailed experiments with many settings on two MoE LLMs, and the figures are well drawn.

**Weaknesses:**

1. Not well motivated: the authors did not state well or give some experimental evidence about the motivation, objective, and benefits of designing a customized PEFT method for MoE. For example, does the proposed method perform better or computationally cheaper on MoE LLMs compared to the widely adopted vanilla LoRA fine-tuning?


2. Lack of Novelty. Actually, this paper is (A+B) type work. It combines some existing designs in MoE, multiple LoRA, MoE-LoRA approaches.

3. Potentially limited generalizability: Considering that there are many different designs for MoE LLMs such as fine-grained grouped experts, shared experts, and for both the single and multi- task scenarios, I am worried about the scalability of the proposed complex designs to different MoE LLMs. The authors need to add experiments in these different scenarios that I mentioned to address this concern.

4. Not Important and Necessary Research Track: Similar to that mentioned in 1, is it necessary and important to design a customized PETF method for MoE LLMs? Why not use a generalized LoRA setting? Are MoE LLMs so widespread that they can be become a new track out of the common LLMs? Similarly, is a customized PEFT needed for some other architectures like SSM LLMs like Mamba, or in case some incremental new architectures like MoE-SSM come up later on, is a new PEFT needed to be designed?

5. Lack of comparisons and experiments under more scenarios: This paper only compares with different of its own designs, and lacks comparisons with more LoRA variants, multi-LoRA, and MoE-LoRA methods. This makes it difficult to show the superiority of this work. In addition, this paper only experiments on two MoE LLMs, and more types and sizes of MoE LLMs need to be considered, such as Deepseek-MoE, Mixtral 8 × 22B, Qwen1.5-MoE-A2.7B, Qwen 2-57B-A14B, DBRX, and Grok-1.


6. Lack of theory. The paper is all empirical analysis lacking theoretical analysis and support, which is not appropriate to appear in machine learning conferences. In fact, the experimental results and analysis are potentially problematic, and it is unconvincing without theoretical support and analysis.


7. The design of the sub-modules is very confusing, complex and unintuitive. It makes it difficult to understand the connections between the modules, not to mention for other community researchers to use them.

**Questions:**

See weaknesses. In fact, I don't think this work meets ICLR's acceptance standards at all, and I'm giving br a preliminary score out of kindness.  I think the authors need to devote lots or even rewrite their paper all over again to completely address my concerns.

---

> ### Author Response · Authors · 2024-11-24
>
> Thank you for your time and feedback on our paper! The responses to your key concerns are below.
>
> ### Weakness 1 & 2 about Motivation and Novelty:
>
> We think you might have missed an important part in our paper. We would like to clarify that it is exactly because the adapters and routing mechanisms have been independently validated that motivated us to propose our unified framework. We aim to contribute to the community by proposing our unified framework **covering ALL specific design dimensions** (functional / compositional strategies) **and possible choices on them that anyone would face when designing PEFT for MoE**.
>
> - There *hasn’t been any previous work applying* ***PEFT to MoE* models**, and we try to not only be the first methods studying this, but to do so systematically.
> - Our work is beyond a simple combination of MoE and PEFT, as we seek to build a framework for all possible design choices, and when doing so we also managed to reveal the intriguing dynamics behind *key memory vectors* in experts and *expert vectors* in routers for both FFN & PEFT experts (as shown in Fig. 3 & 6) which provides insights for fine-tuning MoEs that have rarely been discussed in such detail in previous MoE studies.
> - By combining representative designs under this framework, we proposed our PERFT family. We thoroughly experimented our PERFT variants with an exhaustive list of possible designs, and presented results with empirical observations we identified in our experiments across settings, domains, and models. No MoE fine-tuning has been experimented at this scale before, hence we believe our findings could facilitate the community understanding PEFT and MoE.
>
> Our approach mirrors how He et al. (ICLR’22) [1] have established their unified view of PEFT, where they also explored possible design dimensions and carried out their experiments by combining different design choices. In our work we examined our design choices in a much larger scale and a much more thorough way.
>
> ### Weakness 3 about Limited Generalizability:
>
> We appreciate this concern about generalizability. However, we must point out that rather than exhaustively testing every emerging variant, we believe our current focus better serves the field's current needs.
>
> - Our choice of base models reflects careful consideration of the current research landscape. The reason we implemented our experiments on OLMoE-1B-7B and Mixtral-8$\times$7B is because they are the state-of-the-art MoE models with the best performance/cost ratio (according to MMLU performance reported in OLMoE’s report [2]) under the activated parameter scales of ~1B and ~10B. We believe these two models can be reasonably selected as adequate “standard” representative on the current landscape of mainstream MoE designs.
> - In our work we focus more on introducing our unified framework and by combining design choices to propose PERFT family and empirical findings, following how He et al. (ICLR’22) [1] have established their unified view of PEFT. We emphasize on the possibility of searching within our unified framework that focus on general MoE architectures instead of specific architectures. Instead of experimenting an exhaustive list of existing MoE architectures, our current approach prioritizes depth over breadth, by covering comprehensive design choices within our own framework that is inherently designed for any MoE architecture (See Figure 1 & 2, as long as there is a router and FFN experts), which ensures broader generalizability and applicability and is able to yield more valuable insights for research community.
> - It is also worth noting that regarding compatibility with specific architectures like shared experts, our PERFT variants are inherently designed for seamless integration. Take DeepSeekMoE or QwenMoE as examples - PERFT-R, PERFT-S, and PERFT-D can naturally work in parallel with shared FFN experts as additional shared PEFT experts. In fact, as already detailed in Section 3.1.2 (line 248), their shared expert designs directly motivated our development of shared PEFT experts.
> - Many recent MoE designs are themselves still undergoing validation, and a comprehensive evaluation across all variants would constitute an entirely different research direction. We maintain that this does not consist a critical limitation of our work at this stage, as our current experiments with thorough evaluation on established, state-of-the-art architectures provides sufficient validation of our framework's effectiveness.

---

> > ### Author Response · Authors · 2024-11-24
> >
> > ### Weakness 4 about Necessity of Resarch Track:
> >
> > We would like to highlight several key points that demonstrate why specialized adaptation strategies for MoE is indeed a crucial research direction:
> >
> > 1. We believe that adoption of MoE architecture has de facto become widespread in new transformer designs. As you’ve already mentioned in weakness 5, we've both already seen a large number of remarkable opensource MoE LLMs released recently.  Research suggests even industry-leading models like GPT-4 have also adopted MoE architecture [3]. Among the 17 new text models released during the past year within HuggingFace’s transformers library, nearly half (8) of them are MoE.
> > 2. Regarding it as a research track, among current list of active submissions mentioning "LLM" in their title and metadata, nearly 10% also directly mentions “MoE”.
> > 3. While we agree that not every architectural variation requires specialized fine-tuning methods, with this number of new MoE models released and lots of other studies building on them, we believe it is definitely necessary and beneficial to seriously consider a dedicated research attention on how to efficiently adapting these models for downstream tasks, instead of still directly adopting the baselines that operates in isolation from the MoE routing mechanism.
> >
> > Regarding your concern about the necessity of designing PEFT tailored for MoE in stead of directly applying generic conventional PEFT designed for dense models, we’ve already included these aspects in our paper:
> >
> > 1. It is indeed possible to apply existing vanilla PEFT methods directly to either attention matrices (non-MoE-related modules) or MLP weights (matrices within individual expert modules), and we have already included this in our framework as ***MoE-agnostic*** approaches (Figure 2 b. ③; or Section 3.1.2, line 269), categorized under the compositional strategies.
> > 2. As these methods were originally designed for dense models, when implemented on MoE models they would operate entirely in isolation from the MoE’s router of the model, with no effective interaction nor available to leverage any advantage of the routing dynamics. We take them as baselines through our study.
> > 3. As discussed in Section 3.1, instead of those trivial methods without consideration of the underlying MoE structure, we would like to focus more on methods with better interactions with MoE. Based on our framework, in Section 3.2, we further proposed our PERFT variants that are more tailored for MoE.
> > 4. Our PERFT variants are not only able to outperform conventional PEFT baselines, but also demonstrate intriguing dynamics behind key memory vectors in experts and expert vectors in routers for both FFN & PEFT experts (as shown in Figure 3 & 6), a insight for MoE fine-tuning that have rarely been discussed in such detail among previous studies. We believe these results would be valuable for the research community.
> >
> > Regarding your suggestion concerning PEFT on SSM LLMs, we believe this is irrelevant to our paper and research track. However we’d like to clarify that with the rapid emergence of MoE models and their growing adoption in application, developing efficient, specialized fine-tuning methods is not just beneficial but necessary. Your point about other architectures like SSM or potential MoE-SSM hybrids actually reinforces our position: as architectures evolve, we need systematic frameworks for developing appropriate fine-tuning strategies. Our work follows precedents [1] that provides templates for how to approach such adaptations systematically, rather than relying on generic solutions that may not fully capitalize on architecture-specific advantages.

---

> > > ### Author Response · Authors · 2024-11-24
> > >
> > > ### Weakness 5 about More Experiments:
> > >
> > > Thank you for pointing out the possibility of introducing more experiments.
> > >
> > > 1. Regarding the comparisons, we’ve already taken 2 PEFT solutions in their popular settings (LoRA for QV matrices in attention and parallel adapters) as our baselines. All these baselines serves as “MoE-Agnostic PEFT” methods as the design choices within our unified framework of PEFT for MoE (See Section 3.1.2 and Figure 2).
> > > 2. We admit more experiments could be carried out on more LoRA variants. We are running experiments with more settings on router or/and expert weight matrices, and results will be provided soon in our next revised version.
> > > 3. Regarding testing on more MoE LLMs we have pointed out in our response to your Weakness #3 that rather than exhaustively testing every emerging variant, we believe our current focus better serves the field's current needs. Our choice of base models reflects careful consideration of the current research landscape. The reason we implemented our experiments on OLMoE-1B-7B and Mixtral-8$\times$7B is because they are the state-of-the-art MoE models with the best performance/cost ratio (according to MMLU performance reported in OLMoE’s report [2]) under the activated parameter scales of ~1B and ~10B. We believe these two models can be reasonably selected as adequate “standard” representative on the current landscape of mainstream MoE designs.
> > > 4. Implementing our framework on all mentioned models requires several magnitudes more significant computational budget for scaling and getting observations along the design dimensions.  Without an infinite amount of budget for experiments, we believe that instead of exhaustively experimenting every MoE base models, focusing on more thorough and deeper exploration of design choices within our own frameworks with several selected representative models is better for providing empirical findings that is more useful to the community.
> > >
> > > ### Weakness 6 about Lack of Theory:
> > >
> > > We appreciate your feedback. However, we have to respectfully refrain from your assessment regarding theoretical requirements, and address that from our perspective, ICLR has never historically been, and indeed shall not be, a platform where only works with substantial theoretical analysis and support could be appreciated.
> > >
> > > - Among the list of previously accepted submissions from ICLR’24, only \~24% mentioned “theory / theoretical” in their title and metadata, a ratio similar from those mentioning “framework” (\~28%) or “empirical” (\~20%). This clearly indicates that the machine learning community has long embraced a more inclusive view of meaningful contributions than what you suggested.
> > > - In fact, in rapidly evolving areas like LLMs, it's worth noting that purely empirical technical reports have consistently driven the field's most significant advances. While theoretical understanding certainly has its place, it has typically followed — rather than led — the practical breakthroughs that have actually moved the field forward.
> > > - Your opinion also seems to overlook established precedents in our field. He et al. (ICLR'22) [1], for instance, have successfully introduced frameworks in a similar way, by exploring design dimensions of PEFT through comprehensive empirical analysis. Their methodology - proposing a framework and methodically exploring design combinations - has proven remarkably effective at uncovering valuable insights, much like our approach.
> > >
> > > As clearly articulated in our introduction (lines 90-97), our work delivers concrete contributions through a systematic unified framework (Figure 2), novel insights into previously unexplored dynamics between (Figure 3), and all the empirically-validated design principles across multiple dimension (Section 4.2 & 4.3). Therefore, we maintain our position on this matter, that the practical insights and empirical rigor we provide are enough to make meaningful contributions to the understanding of MoE and PEFT, and align precisely with what the community has consistently valued.

---

> > > > ### Author Response · Authors · 2024-11-24
> > > >
> > > > ### Weakness 7 about Better Design:
> > > >
> > > > We’ll try our best to improve our articulation in the next revised version! We think you might have missed our Figure 1 & 2, which have already set a very clear illustration for facilitating understanding how models are designed and implemented. These figures have been specifically crafted to facilitate understanding of the model architecture and module interactions.
> > > >
> > > > [1] He, J., Zhou, C., Ma, X., Berg-Kirkpatrick, T., & Neubig, G. (2021). Towards a unified view of parameter-efficient transfer learning. *arXiv preprint arXiv:2110.04366*.
> > > >
> > > > [2] Muennighoff, N., Soldaini, L., Groeneveld, D., Lo, K., Morrison, J., Min, S., ... & Hajishirzi, H. (2024). OLMoE: Open Mixture-of-Experts Language Models. *arXiv preprint arXiv:2409.02060*.
> > > >
> > > > [3] Patel, D. & Wong, G. (2023). GPT-4 Architecture, Infrastructure, Training Dataset, Costs, Vision, MoE // Demystifying GPT-4: The engineering tradeoffs that led OpenAI to their architecture. https://semianalysis.com/2023/07/10/gpt-4-architecture-infrastructure/

---

> > > > > ### Comment · Reviewer_SUvZ · 2024-11-26
> > > > >
> > > > > I have read the replies. Thanks.

---

> > > > > > ### Comment · Reviewer_SUvZ · 2024-11-28
> > > > > >
> > > > > > The author's reply cleared up most of my doubts. I think the author has some new discoveries and designs on Moe's fine-tuning and gives solid experiments. Considering these factors, I decided to improve my score. Thank you for the author's reply.

---

### Official Review · Reviewer_hrjf · 2024-11-04

**Soundness:** 2
**Presentation:** 2
**Contribution:** 1
**Rating:** 3
**Confidence:** 4

**Summary:**

The paper introduces "Parameter-Efficient Routed Fine-Tuning" (PERFT), a framework for efficiently finetuning Mixture-of-Experts models. PERFT offers a set of strategies tailored to MoE, and claims to reduce the resource intensity traditionally required for finetuning such large models.

**Strengths:**

The work tries to be one of the first methods to introduces Parameter-Efficient Fine-Tuning (PEFT) methods which are applicable directly and only to Mixture-of-Experts models.

**Weaknesses:**

Line 31 Mixture of Experts models do not significantly or necessarily reduce computation costs since routing methods may be complex and may not optimize and generalize trivially, and also significantly increase memory costs for both training and inference. This assertion is not entirely correct and forms an inadequate motivation for this work.

Line 38 "Motivation is unclear"
The usage of terms 'dense models' is not explained at the outset but is used all the time. Additionally, the term 'Peft strategies .. tailored for MoE models' remains ambiguous and unmotivated. Does it refer to making specific peft models for mixture of experts models? Why do we need this separation from normal peft models?

Line 47 "...our framework focuses on the core principles and unique
challenges of MoE architecture." - This statement is not followed by any explanation of what these principles and challenges are

The application shown is quite limited (to reasoning tasks). It remains unclear to see if the current methodology is applicable to other tasks and domains as well (Even in simpler tasks like image segmentation or classification), especially in harder tasks like text to image generation and complex vision-language architectures.

MoE models often suffer with training instability[3] and load balancing issues[1, 2] (i.e. uneven load distribution among experts), therefore diverse testing (even on smaller models) is necessary. The paper doesn't provide any analysis of whether this peft methodology acerbates or ameliorates these problems and if it does not improves on these issues, then why not simply apply normal peft methods to individual experts?

"Novelty" I had some difficulty in understanding the novelty of the method introduced. The adapters and the routing mechanisms seem conventional and its unclear what in terms of novelty has been presented in this work (method, theoretical proofs and novel empirical results).

"Increased Complexity" Since, a new routing methodology is to be learnt between the peft modules in addition to the routing between the original experts, the motivation to do so is not explained in the work. Routing mechanism can be infamously hard to optimize and may lack ability to generalize to different tasks. It remains unclear why have this increased complexity in the methodology when a peft model can be learnt for an individual expert and these modified experts be combined in a new mixture. Also, is there any computational and memory advantage (both during training and inference) for using this work's method over learning individual peft for each expert model?

"Readability" The paper is a bit hard to read given often the method and assertions are not well motivated and explained. Terms which may be common to this subfield are not explained initially or in the appendix and are used throughout the paper. For e.g. line 201 - the paragraph "Routing among PEFT Experts" does not explain whether it is important or necessary or not and neither provides any theoretical insight or empirical evidence and just says "This aspect highlights the profound dynamics between routers and experts in MoE and PEFT modules". This does not make much sense for a reader.

[1] Chen et. al. - Towards Understanding the Mixture-of-Experts Layer in Deep Learning

[2] Shazeer, N., Mirhoseini, A., Maziarz, K., et al. (2017). Outrageously Large Neural Networks: The Sparsely-Gated Mixture-of-Experts Layer

[3] Lepikhin, D., Lee, H., Xu, Y., et al. (2021). GShard: Scaling Giant Models with Conditional Computation.

**Questions:**

Please refer to the questions in the weakness section. Improving the readability by stressing on the motivation, novelty and intuition of the method would make this work much better.

---

> ### Author Response · Authors · 2024-11-24
>
> Thank you for your time and feedback on our paper! We really appreciate your detailed identification of strengths and weaknesses of our work! Please find our response to your comments below.
>
> ### Weakness Line 31:
>
> Thank you for your opinion. Regarding your concerns on computation costs, we believe there may be a misunderstanding about how MoE achieves its computational advantages over conventional designs. The efficiency benefits of MoE should be considered under controlled comparisons:
>
> - **Active parameter efficiency.** When compared to dense LMs with equivalent **active** parameters, MoE would “train ~2$\times$ faster” and have similar inference cost with “significantly” better performance;
> - **Total parameter efficiency.** When compared to dense LMs with equivalent **total** parameters, MoE would both train and inference faster, while remaining a “competitive” performance.
>
> These cited efficiency characteristics are well-documented in [1, page 3; 8], and our framing aligns with the established understanding in the field.
>
> As for your concerns on challenges brought by routers, we fully agree that the complexity of MoE's routing mechanisms, including load-balancing and stability considerations, does indeed add significant challenges to training of them. However, instead of viewing it as an inadequate motivation, we actually consider this complexity as further strengthening our motivation — it underscores precisely why MoE models require dedicated PEFT strategies rather than direct application of existing vanilla PEFT methods.
>
> We appreciate you bringing this to our attention and will revise our manuscript to better reflect these nuances.
>
>
> ### Weakness Line 38:
>
> We apologize for the lack of definition for “dense” models. We used this term as its widely-adopted usage in previous literatures [1-4], which refers to the design of FFNs in conventional transformers, in contrast to MoE as “sparse” architectures with sparsely activated submodules.
>
> Regarding your concern about the necessity of designing PEFT tailored for MoE in stead of directly applying conventional PEFT designed for dense models:
>
> 1. It is indeed possible to apply existing vanilla PEFT methods directly to either attention matrices (non-MoE-related modules) or MLP weights (matrices within individual expert modules), and we have already included this in our framework as ***MoE-agnostic*** approaches (Figure 2 b. ③; or Section 3.1.2, line 269), categorized under the compositional strategies.
> 2. As these methods were originally designed for dense models, when implemented on MoE models they would operate entirely in isolation from the MoE’s router of the model, with no effective interaction nor available to leverage any advantage of the routing dynamics. We take them as baselines through our study.
> 3. As discussed in Section 3.1, **instead of those trivial methods without consideration of the underlying MoE, we would like to focus on methods that more actively interacts with the MoE mechanism**. Based on our framework, in Section 3.2, we further proposed our PERFT variants that are more tailored for MoE.
> 4. Our PERFT variants are not only able to **outperform** conventional PEFT baselines, but also demonstrate **intriguing dynamics** behind key memory vectors in experts and expert vectors in routers for both FFN & PEFT experts (as shown in Figure 3 & 6), a insight for MoE fine-tuning that have rarely been discussed in such detail among previous studies. We believe these results would be valuable for the research community.

---

> > ### Author Response · Authors · 2024-11-24
> >
> > ### Weakness Line 47:
> >
> > Thank you for highlighting this gap in our explanation. Let us clarify what we mean by "core principles and unique challenges" of MoE and how they inform the design of our framework and PERFT family.
> >
> > - **Leveraging dense transformer’s characteristics have helped MoE achieve better performance & efficiency.** The introduction of MoE to transformers was originally motivated by the observation of inherent sparsity within dense model’s FFN layers. By leveraging this sparsity through dynamic routing among experts, MoE manages to achieve increased model capacity without proportional computational costs. We refer this utilization of sparsity as MoE’s core principles and its solution to address the challenges for better scaling-up transformers.
> > - **Following this logic and taking MoE’s characteristics into account, we aim at designing better PEFT approach for MoE.** We notice that conventional PEFT approaches agnostically treat the MoE base model just as any other dense models, without considering any unique sparse and dynamic nature behind them. This leave us room for designing PEFT methods that also leverage these characteristics. Also intuitively, the original MoE modules that sparsely activates FFN experts, which encodes different knowledge, should require the introduced PEFT module to actively interact with its routing pattern. This calls for designing ideal PEFT methods for MoE that better capture their routing patterns.
> > - Our framework directly taking the sparsity of MoE into account and addresses these challenges, in both Section 3.1.1 Functional Strategy and Section 3.1.2 Compositional Strategy. We demonstrated in Figure 3 & 6 the intriguing dynamics behind how routers for introduced shared PEFT experts would manage their interaction with the original routing pattern and FFN experts, and in line 215 & 266 how embedded PEFT modules can be viewed as equivalent shared PEFT modules, with directly using the same routing pattern instead of learning a new one. All these considerations leads to PERFT-R and PERFT-E’s superior performance against MoE-agnostic baselines in our experiments, validating our framework’s effectiveness in addressing the unique sparse characteristics of MoE.
> >
> > We appreciate you bringing this to our attention, and will revise the related sections to better articulate these points.
> >
> > ### Weakness about Limited Application:
> >
> > We must respectfully point out that this comment appears to reflect some misunderstanding about the current state of MoE research and development.
> >
> > - Our work focuses on **PEFT methods** for MoE models, which necessarily requires the **existence of pretrained MoE architectures in the target domain**. At present, the mainstream open-source MoE models are predominantly LLMs pretrained (and fine-tuned) for tasks purely within language domain. Your suggestion about applying our methods to visual and multimodal domains overlooks the crucial prerequisite of an applicable pretrained MoE model and a downstream dataset for fine-tuning in these domains. **This is not a limitation of our methodology, but rather reflects the current landscape of available MoE models and downstream tasks.**
> > - As transferring across domains (LLM → VLM) always requires full fine-tuning instead of PEFT [5,6], for implementing your suggested experiments, we would first need to pretrain (or continued-pretraining) MoE models by ourselves in these alternative domains, an endeavor that itself would constitute an independent work [6] and belongs to an entirely different research direction that falls well outside the scope of our work on PEFT methods.
> >
> > ### Weakness about Training Stability and Load Balancing Issues:
> >
> > We would like to point out that these issues are irrelevant to PEFT.
> > 1. To our knowledge, currently the standard solution for these issues is by incorporating ***auxiliary loss*** at the LM head, rather than introducing additional architectural modifications like LoRA matrices or adapter modules within transformer blocks.
> > 2. The issues you raise are **typically more concerned during pretraining rather than the fine-tuning stage**. Both problems are less critical during fine-tuning when working with already well-trained base models.
> > 3. In our experiments we directly use 2 well-pretrained MoE models as our base models, and during our fine-tuning, we have incorporated the exactly same load-balancing loss & z-loss used during its pretraining, which ensures consistent training conditions for fair comparison.
> > - Your suggestion about applying PEFT to individual experts is actually already covered in our framework under Section 3.1.2 Embedded PEFT Experts. We demonstrated the structural equivalence between embedded and shared PEFT experts when the number of Top k/M meets the original FFN experts’ settings. For another potential baseline featuring LoRA at matrices within each FFN expert, we’ll include a series of additional experiment in our appendices in the next revised version.

---

> > > ### Author Response · Authors · 2024-11-24
> > >
> > > ### Weakness about Novelty:
> > >
> > > We would like to clarify that it is exactly because the adapters and routing mechanisms have been independently validated that motivated us to propose our unified framework. We aim to contribute to the community by proposing our unified framework **covering ALL specific design dimensions** (functional / compositional strategies) **and possible choices on them that anyone would face when designing PEFT for MoE**.
> > >
> > > - There *hasn’t been any previous work applying* ***PEFT to MoE* models**, and we try to not only be the first methods studying this, but to do so systematically.
> > > - Our work is beyond a simple combination of MoE and PEFT, as we seek to build a framework for all possible design choices, and when doing so we also managed to reveal the intriguing dynamics behind *key memory vectors* in experts and *expert vectors* in routers for both FFN & PEFT experts (as shown in Fig. 3 & 6) which provides insights for fine-tuning MoEs that have rarely been discussed in such detail in previous MoE studies.
> > > - By combining representative designs under this framework, we proposed our PERFT family. We thoroughly experimented our PERFT variants with an exhaustive list of possible designs, and presented results with empirical observations we identified in our experiments across settings, domains, and models. No MoE fine-tuning has been experimented at this scale before, hence we believe our findings could facilitate the community understanding PEFT and MoE.
> > >
> > > Our approach mirrors how He et al. (ICLR’22) [7] have established their unified view of PEFT, where they also explored possible design dimensions and carried out their experiments by combining different design choices. In our work we examined our design choices in a much larger scale and a much more thorough way.
> > >
> > > ### Weakness about Increased Complexity:
> > >
> > > The reason that motivated us to propose and study PERFT-R with a routing among introduced PEFT modules in addition to the original routing between the FFN experts is based on several observations:
> > >
> > > 1. In Section 3.1.1 Functional Strategy, our framework has identified this design of introducing routing among PEFT experts as a possible choice in designing PEFT for MoE models.
> > > 2. Recent works on PEFT for dense models (Section 2.2 PEFT with MoE-like Structures) have studied numerous similar MoE-like PEFT methods which has shown improvements, suggesting that in MoE models the design of routing among PEFT experts might also be a highly competitive choice to consider.
> > > 3. We demonstrated a potential of the benefits from the dynamics between key-memory vectors of FFN experts and expert vectors in routers for both PEFT experts and FFN experts in our Figure 3.
> > > 4. In our experiments we verified that PERFT-R with its routing among PEFT modules is, as expected, generally able to achieve the best performance across domains and different activated parameter settings.
> > >
> > > You mentioned that “a peft model can be learnt for an individual expert and these modified experts be combined in a new mixture”, and this is analyzed in Section 3.2 as the equivalence between PERFT-E and PERFT-R, by which we can view PERFT-E as a simplified special case of  PERFT-R that utilizes the original router’s routing patterns instead of learning a independent router (See Eq. (8) at line 288) . This can only be achieved when the number of introduced PEFT experts $M$ matches the number of FFN experts $N$, and the activated PEFT experts matches the number of activated FFN experts as well, which restricts its flexibility as leaving the bottleneck size $D_B$ as the only scaling factor available. In contrast, PERFT-R allows arbitrary settings for selecting Top $K$/$M$ PEFT experts to activate. We compared and analyzed the experiment results of PERFT-R and PERFT-E in Section 4.3 line 479-514.
> > >
> > > The difference in the **computational cost** during training and inference is controlled in our experiments via “# Act.” and “% Act.” in Table 1, and “activated parameter efficiency” in Fig. 4 & 5. We compared different methods with similar scale of activated parameters, which directly translate to FLOPs incurred during both fine-tuning and inference.
> > >
> > > For the **memory cost**, as all our methods are within the scope of Parameter-Efficient Fine-Tuning methods, the total amount of parameters introduced should be marginal in comparison to the total amount of parameters in the MoE base model. In our experiments we kept our largest experiments under the scale of <1%, which means no special consideration of memory advantage is needed for detailed discussion.

---

> > > > ### Author Response · Authors · 2024-11-24
> > > >
> > > > (continued)
> > > >
> > > > Beyond computational and memory costs, there actually is another difference of PERFT-R and PERFT-E in training that we emphasized in our paper, that is their training dynamics. It is intuitive that as the number of PEFT experts increases, learning a new router in PERFT-R would become less favorable and could harm the performance. This requires balancing between training stability that PERFT-E offers in its direct utilization of pretrained router and the flexibility that PERFT-R offers in learning a new router which could support different Top $K$/$M$ settings.  We demonstrate in Fig. 6 a post-hoc visualization of learned expert vectors in PERFT-E and PERFT-R with different settings.
> > > >
> > > > ### Weakness about Readability:
> > > >
> > > > Thank you for pointing this out. We would like to clarify your concerns regarding the motivation and explanation of our design.
> > > >
> > > > The introduction of routing among PEFT experts is well-motivated by several key observations:
> > > >
> > > > 1. In Section 3.1.1 Functional Strategy, our framework has identified and supported this potential design of introducing routing among PEFT experts.
> > > > 2. Recent works on PEFT for dense models (Section 2.2 PEFT with MoE-like Structures) have studied numerous similar MoE-like PEFT methods which has shown improvements, suggesting that in MoE models the design of routing among PEFT experts might also be a highly competitive choice to consider.
> > > > 3. We demonstrated a potential of the benefits from the dynamics between key-memory vectors of FFN experts and expert vectors in routers for both PEFT experts and FFN experts in our Fig. 3.
> > > > 4. In our experiments we verified that PERFT-R with its routing among PEFT modules is, as expected, generally able to achieve the best performance across domains and different activated parameter settings.
> > > >
> > > > [1] Muennighoff, N., Soldaini, L., Groeneveld, D., Lo, K., Morrison, J., Min, S., ... & Hajishirzi, H. (2024). OLMoE: Open Mixture-of-Experts Language Models. *arXiv preprint arXiv:2409.02060*.
> > > >
> > > > [2] Dai, D., Deng, C., Zhao, C., Xu, R. X., Gao, H., Chen, D., ... & Liang, W. (2024). DeepSeekMoe: Towards Ultimate Expert Specialization in Mixture-of-Experts Language Models. *arXiv preprint arXiv:2401.06066*.
> > > >
> > > > [3] Rajbhandari, S., Li, C., Yao, Z., Zhang, M., Aminabadi, R. Y., Awan, A. A., ... & He, Y. (2022, June). Deepspeed-moe: Advancing mixture-of-experts inference and training to power next-generation ai scale. In *International conference on machine learning* (pp. 18332-18346). PMLR.
> > > >
> > > > [4] Lepikhin, D., Lee, H., Xu, Y., Chen, D., Firat, O., Huang, Y., ... & Chen, Z. (2020). Gshard: Scaling giant models with conditional computation and automatic sharding. *arXiv preprint arXiv:2006.16668*.
> > > >
> > > > [5] Liu, H., Li, C., Wu, Q., & Lee, Y. J. (2024). Visual instruction tuning. *Advances in neural information processing systems*, *36*.
> > > >
> > > > [6] Lin, B., Tang, Z., Ye, Y., Cui, J., Zhu, B., Jin, P., … & Yuan, L. (2024). MoE-LLaVA: Mixture of Experts for Large Vision-Language Models. *arXiv preprint arXiv:2401.15947*.
> > > >
> > > > [7] He, J., Zhou, C., Ma, X., Berg-Kirkpatrick, T., & Neubig, G. (2021). Towards a unified view of parameter-efficient transfer learning. *arXiv preprint arXiv:2110.04366*.
> > > >
> > > > [8] Liu, L., Kim, Y. J., Wang, S., Liang, C., Shen, Y., Cheng, H., ... & Chen, W. (2024). GRIN: GRadient-INformed MoE. *arXiv preprint arXiv:2409.12136.*

---

> > ### Comment · Reviewer_hrjf · 2024-11-24
> > **Response to Author Rebuttal.**
> >
> > Line 31. My question was regarding the overly broad and ambiguous claim made regarding MoE models in line 31 without any evidence or analysis. Please mathematically define what you mean by "controlled comparisons" - this is again an overtly broad and ambiguous definition.
> > Are the claims made regarding efficiency applicable to all MoE models and across all domains? IT seems so because the statements made do not provide a narrow field of application.
> >
> > Please note that [1] is not only **not a peer-reviewed publication** but appears to be under review at ICLR itself. This citation currently, unfortunately, does not provide evidence for your paper's motivations.
> >
> > Line 38. Thank you for the reply. However, although your response is long, its till doesn't provide a strong motivation for your method. There are significant number of MoE models which utilize LoRAs in addition to "dense" models. Why do we want a PeFT method which interacts with the MoE parameters exclusively? Is there any motivation and results or theory for this? Any hypothesis? What are the "intriguing dynamics" and what are the insights they bring that motivates your method?
> >
> > Line 47. The concept of MoE predate transformer architectures. Are you referring to a specific MoE archicture? Why does then paper make broadly applying claims and assumptions regarding MoE. This explanation seems very vague. Not all MoE models have any kinds of sparsity. As far as i am aware - there are no MoE papers which claim that there is any sparsity property which exists for the expert models. in case your paper refers to a very specific model and paper, I'd recommend rewritting to fix the broadly applicable claims.
> >
> > Please read my comment again. I'm not suggesting only VLMs - even simpler tasks like image segmentation and classification for which a large number of MoE methods are available are very applicable and doable for you method. in the current state, the experiments do not provide a strong support for the method especially since this is an empirically based work.
> >
> > PeFT method are not applicable to training instability and load balancing issues of MoE models but yours is since you are designing a PeFT model for MoEs. So, what effect does it bring about in regards to these issues?

---

> > > ### Author Response · Authors · 2024-12-01
> > >
> > > Dear Reviewer **hrjf**,
> > >
> > > Thank you for your feedback. We appreciate it and would like to further clarify some points in our revised paper to help address your concerns.
> > >
> > > - ### Regrading the previous Line 31:
> > >
> > >     Regarding your original claim about MoE model’s efficiency
> > >
> > >     > *Mixture of Experts models do not significantly or necessarily reduce computation costs since routing methods may be complex and may not optimize and generalize trivially, and also significantly increase memory costs for both training and inference.*
> > >     >
> > >
> > >     **We’ve already removed these parts in our revised version**, since they **only served as background introduction rather than an integral part in our framework, and it is not us who made these claims.**  This part is currently modified as
> > >
> > >     > *With so many new MoE LLMs available, how to effectively fine-tune them for downstream tasks has become an area of considerable value.*
> > >     >
> > >
> > >     We still like to point out that your previous opinion in this matter seems in the contrary to the current understandings in both academia and industry. **The claim of efficiency we previously cited has been explicitly demonstrated across numerous works [1-9]**, including technical reports of **almost EVERY MoE LLMs ever released**, published conference papers, and well-validated works with >1k citations, even in early studies you provided, like Sparsely-Gate MoE [1] and GShard [2]:
> > >
> > >     > Sparsely-Gate MoE [1]:
> > >     *For all of our MoE models, the **floating point operations involved** in the experts represent between **37% and 46% of the total**. For our baseline models wtih no MoE, observed computational efficiency ranged from 1.07-1.29 TFLOPS/GPU … Our **highest-computation MoE model was more efficient** at 1.56 TFLOPS/GPU, likely due to the larger matrices. These numbers represent a significant fraction of the theoretical maximum of 4.29 TFLOPS/GPU claimed by NVIDIA.*
> > >     >
> > >
> > >     > GShard [2]:
> > >     *We … demonstrated a 600B parameter multilingual neural machine translation model can **efficiently be trained** in 4 days **achieving superior performance and quality compared to prior art** when translating 100 languages to English with a single model. In addition to the far better translation quality, MoE Transformer models trained with GShard also **excel at training efficiency**, with a training cost of 22 TPU v3 core years compared to 29 TPU years used for training all 100 bilingual Transformer baseline models.*
> > >     >
> > >
> > >     > Mixtral [3]:
> > >     *Mixtral only uses 13B active parameters for each token. **With 5x lower active
> > >     parameters,** Mixtral is able to **outperform Llama 2-70B** across most categories … The **memory costs** for serving Mixtral are **proportional to its sparse parameter count**,
> > >     47B, which is still smaller than Llama 2 70B.*
> > >     >
> > >
> > >     > DeepSeekMoE [4]:
> > >     *Evaluation results reveal that **with only about 40% of computations**, DeepSeekMoE-16B achieves comparable performance with DeepSeek-7B, … DeepSeekMoE-16B **consistently outperforms models with a similar number of activated parameters** by a large margin, and achieves **comparable performance with LLaMA2-7B, which has approximately 2.5 times the activated parameters.***
> > >     >
> > >
> > >     > QwenMoE [5]:
> > >     *… we have observed a remarkable **reduction of 75% in training costs** when using Qwen1.5-MoE-A2.7B in comparison to Qwen1.5-7B … Qwen1.5-MoE-A2.7B model exhibits an impressive improvement in speed, being **approximately 1.74 times faster** compared to the Qwen1.5-7B model. This acceleration is primarily attributed to the fact that the MoE architecture activates a notably smaller portion of its total parameters, thereby **reducing computational demands.***
> > >     >
> > >
> > >     > Grok-1 [6]:
> > >     *… Grok-1 displayed strong results, **surpassing all other models in its compute class**, including ChatGPT-3.5 and Inflection-1. It is only surpassed by models that were trained with a significantly larger amount of training data and compute resources like GPT-4. This showcases the rapid progress we are making at xAI in **training LLMs with exceptional efficiency**.*
> > >     >
> > >
> > >     > JetMoE [7]:
> > >     *This report introduces JetMoE-8B, a new LLM **trained with less than $0.1 million**, … JetMoE-8B outperforms the Llama2-7B model, and JetMoE-8B-Chat outperforms the Llama2-13B-Chat model, demonstrating that LLM **training can be much more cost-effective than generally thought**. In addition, JetMoE-8B has 8B parameters while only activating 2B for each input token, **reducing inference computation by about 70%** compared to Llama2-7B.*
> > >     >

---

> ### Author Response · Authors · 2024-12-01
>
> (continued)
> > phi-3.5-MoE [8]:
> *The phi-3.5-MoE adopts an Mixture-of-Experts (MoE) architecture to selectively activate parts of modules on specific inputs to **improve the model efficiency**.*
> >
>
> > GRIN [9]:
> *GRIN MoE achieves an average score of 79.58 in Table 2, outperforming 7B dense model (average score of 75.74) and matching the 14B dense model (average score of 78.46) trained on the same data … we are able to achieve **over 80% relative training efficiency improvement, compared to a dense model with the same active parameters**, for GRIN MoE training.*
> >
>
> We believe these details should demonstrate the current understanding shared among different studies, and would be sufficient to serve as a reply to your previous opinion that “Mixture of Experts models do not significantly or necessarily reduce computation costs”. Please let us know if this would help you in better comprehending this matter.
>
> - ### Regrading the previous Line 38:
>
>     Thank you for your reply. We’ll separately respond to each one of your concerns.
>
>     > *There are significant number of MoE models which utilize LoRAs in addition to "dense" models.*
>     >
>
>     We feel that you might have used the term “MoE models” incorrectly here, since it contradicts to the “’dense’ models” you mentioned later and brings confusion.
>
>     - If the “significant number of MoE models” you mentioned are the actual MoE LLMs like Mixtral or OLMoE we used in our paper, then these models are MoE models themselves, and there is no “‘dense’ models” in the first place to “utilize LoRAs in addition to” as you suggested later.
>     - If the “significant number of MoE models” refers to *PEFT methods with MoE-like Structures*, we already mentioned in our paper [line 146-153] and in our previous response. **These methods only studied how to adapt Dense LLMs, and with so many new MoE LLMs recently becomes available, it is necessary to study how to effectively fine-tune them for downstream tasks.**
>
>     > *Why do we want a PeFT method which interacts with the MoE parameters exclusively? Is there any motivation and results or theory for this? Any hypothesis?*
>     >
>
>     We **already mentioned these in our paper and our previous response:**
>
>     1. With so many new MoE LLMs recently become available, we **need to study how to efficiently fine-tune them**.
>     2. However, **all the current PEFT methods are designed on dense models** and are ***MoE-agnostic***, operating in isolation from the underlying MoE architecture and ignoring the dynamics in the router.
>     3. Meanwhile, **no previous research has studied how to PEFT an MoE model,** so we need to systematically explore possible design dimensions from scratch that help address these concerns. Therefore, we **identified and filled in this research gap** by proposing our unified framework considering strategies covering all the design choices.
>     4. Among these choices, we have achieved to develop **methods that more actively interacts with the MoE mechanism** [The PERFT family, Section 3.2] with significantly **improved performance** [Section 4.2] and **the dynamics** [Section 3.1, Figure 3 &  Section 4.3, Figure 6] behind key memory vectors in experts and expert vectors in routers for both FFN & PEFT experts.
>
>     You may find the related part in our paper and our previous response.
>
>     > *What are the "intriguing dynamics" and what are the insights they bring that motivates your method?*
>     >
>
>     This consists a very important part in our paper! We uncover the **previously underexplored dynamics between vectors in FFN experts, FFN expert router and PEFT expert router (visualized in Figure 3)**, which provides a novel and intriguing understanding of how the vectors in the router for PEFT experts can interact with the vectors in the original router and the FFN experts [line 209-238]. This brings the count of possible routing patterns to combinatorial numbers, which create much more exploration of hidden state patterns to be stored as vectors in routers. This insight suggests that **routing mechanisms can be leveraged not just for FFN expert selection, but also as a important integral part of the PEFT process itself, an insight that facilitated our design and could also have significant implications for future MoE and PEFT research.**

---

> > ### Author Response · Authors · 2024-12-01
> >
> > - ### Regrading the previous Line 47:
> >
> >     Thank you for the suggestion. In our current version, we’ve revised this into
> >
> >     > *Different from previous PEFT solutions that operate in isolation from the underlying MoE architecture, our framework is designed **closely around the unique routing mechanisms among experts in MoE models.***
> >     >
> >
> >     We actually focus closely on the **routing mechanisms within MoE models**, which is the key design element (**Mixture** of Experts) that set them apart from other architectures.
> >
> >     Our original version was only following the previous research route of MoE, as the principle of MoE came from designing sparsely activated expert modules as a solution [1,2, also see Section 2.1 Mixture-of-Experts in our paper]. We acknowledge your concerns, and we have never implied that all kinds of sparsity should exist in MoE models.
> >
> > - ### Regrading domains other than LLM:
> >
> >     We appreciate your suggestion and have articulated our scope more precisely. We have revised our title, abstract and introduction to **set our focus within the domain of MoE Large Language Models.** The core motivation for this paper is that with these vast amount of MoE LLMs recently becomes available, we need to study how to efficiently fine-tune them for downstream tasks. Its extendibility to vision and multimodal domains is beyond the focus and scope of our current stage.
> >
> > - ### Regrading training stability and load balancing:
> >
> >     We find your response a little bit ambiguous. As you suggest, PEFT method are not applicable to training instability and load balancing issues, and this is also true when studying our method’s effect to these issues, since the PEFT method we designed for MoE still falls within the broader domain of PEFT methods.
> >
> >     As we’ve already clarified in our previous response, these issues belongs to fundamentally different research directions:
> >
> >     - These issues are **mostly considered during pretraining**.
> >         - When we apply PEFT methods, including PERFT, we're working with already-pretrained MoE models where these foundational issues have been stabilized through appropriate techniques. In typical PEFT practices, these pretrained weights are frozen, and only the additionally introduced parameters, with <1% amount of the total size, are trainable.
> >         - In addition, the entire PEFT process involves only <1B of tokens (in our case, ~100M tokens for 3 epochs), in contrast to the Trillions of tokens used during pretraining stage.
> >
> >         Altogether, the effects of these issues become negligible during PEFT, hence typically fall outside the scope of PEFT studies.
> >
> >     - The established approach to addressing these issues in MoE architectures relies on **auxiliary losses** that modify the loss landscape to help better guide the optimizers, rather than introducing additional architectural designs. These auxiliary losses have already been widely adopted as **well-validated standard solutions** across various MoE models [3,4,5,7,8,9] for tackling these issues. In our experiments, we directly applied these established solutions [Section 4.1 line 323-329], rather than attempting to reimagine it from scratch as any novel fundamental architectural solution.
> >
> > We appreciate your time and effort, and we hope these clarifications address your reply and better demonstrate the value of our contributions. Please let us know if you have any additional concerns.
> >
> > **References**
> >
> > [1] Shazeer, Noam, et al. "Outrageously Large Neural Networks: The Sparsely-Gated Mixture-of-Experts Layer." *International Conference on Learning Representations* (2016).
> >
> > [2] Lepikhin, Dmitry, et al. "GShard: Scaling Giant Models with Conditional Computation and Automatic Sharding." *International Conference on Learning Representations* (2021).
> >
> > [3] Jiang, Albert Q., et al. "Mixtral of experts." arXiv preprint arXiv:2401.04088 (2024).
> >
> > [4] Dai, Damai, et al. "DeepSeekMoE: Towards Ultimate Expert Specialization in Mixture-of-Experts Language Models." *CoRR* (2024).
> >
> > [5] Qwen Team.”Qwen1.5-MoE: Matching 7B Model Performance with 1/3 Activated Parameters”. https://qwenlm.github.io/blog/qwen-moe/ (2024).
> >
> > [6] Grok. “Announcing Grok.” https://x.ai/blog/grok (2024).
> >
> > [7] Shen, Yikang, et al. "Jetmoe: Reaching llama2 performance with 0.1 m dollars." arXiv preprint arXiv:2404.07413 (2024).
> >
> > [8] Abdin, Marah, et al. "Phi-3 technical report: A highly capable language model locally on your phone." arXiv preprint arXiv:2404.14219 (2024).
> >
> > [9] Liu, Liyuan, et al. "GRIN: GRadient-INformed MoE." arXiv preprint arXiv:2409.12136 (2024).

---

### Official Review · Reviewer_T3WA · 2024-11-04

**Soundness:** 2
**Presentation:** 3
**Contribution:** 2
**Rating:** 5
**Confidence:** 4

**Summary:**

In this paper, the authors proposed Parameter-Efficient Routed Fine-Tuning (PERFT), a framework of Parameter-Efficient Fine-Tuning (PEFT) strategies for MoE models. Multiple experiments on different datasets and models showed improved performance over baseline methods.

**Strengths:**

1. The proposed framework makes sense and is technically sound to me.
2. The authors conducted several experiments on multiple tasks to show improved performance.
3. Writing is good and easy to follow.

**Weaknesses:**

1. The proposed framework is more of a combination of straightforward implementations of different PEFT strategies. I did not find much insight in the current draft. Even though there are several design choices provided in the current draft, these are more of a permutation of different ways adding PEFT to MoE architecture. Both PEFT and MoE have been validated to help and there is no surprise or insight by simply combining them together.
2. The experimental results are rather inconclusive and task dependent. I am not sure how beneficial it would be to the research community. For example, for PERFT-E  and PERFT-R in Table 1, I am not sure which one is better than the other. What's more, as the authors mentioned in line 418, "These divergent patterns reveal that the optimal configuration appears to be
task-dependent". Without a conclusive design choice or rule of thumb design principal, I am not sure how applicable it would be.
3. Figure 4 is very difficult to interpret. A better design and legend is needed. There are over 10 different color and lines in the figure and without a legend, it is very difficult to keep track of them.

**Questions:**

Overall, I think the proposed framework is more of a vanilla combination of different strategies without much insight. Please refer to the weakness section for details and prepare rebuttal accordingly.

---

> ### Author Response · Authors · 2024-11-24
>
> Thank you for your time and valuable feedback! We really appreciate your identification of strengths and weaknesses of our work. Please find our response to your key concerns below.
>
> ### Weakness 1 about Novelty and Insights:
>
> We would like to clarify that we aim to contribute to the community exactly by proposing our **unified framework covering ALL specific design dimensions** (functional / compositional strategies) and possible choices on them **that anyone would face when designing PEFT for MoE LLMs**.
>
> - There *hasn’t been any previous work applying* ***PEFT to MoE* models**, and we try to not only be the first methods studying this (as suggested by reviewer hrjf), but to do so systematically.
> - Our work is beyond a simple combination of MoE and PEFT, as we seek to build a framework for all possible design choices. And when doing so, we also managed to reveal the intriguing dynamics behind *key memory vectors* in experts and *expert vectors* in routers for both FFN & PEFT experts (as shown in Fig. 3 & 6) which provides insights for fine-tuning MoEs that have rarely been discussed in such detail in previous MoE studies.
> - By exploring representative designs under this framework, we proposed our PERFT family. We thoroughly experimented our PERFT variants with an exhaustive list of possible designs, and presented results with empirical observations we identified in our experiments across settings, domains, and models. No MoE fine-tuning has been experimented at this scale before (with in total over 160 settings being independently parameter-efficiently fine-tuned and >1100 independent evaluation scores reported).
> - We identified a number of consistent and practical insights that generalize across various parameter settings, tasks, and models,  as reported in Section 4.2 & 3. We believe these findings could facilitate the community better applicate PEFT and MoE.
>
> Our approach mirrors how He et al. (ICLR’22) [1] have established their unified view of PEFT, where they also explored possible design dimensions and carried out their experiments by combining different design choices. In our work we examined our design choices in a much larger scale and a much more thorough way.
>
> ### Weakness 2 about Interpreting Experiment Results:
>
> We think you might have missed an important part in our paper. We agree that conclusive design choices and rule of thumb design principals are beneficial to the research community, and that is exactly the reason why we have already provided a number of empirical observations along with experiment results in Section 4.2 & 4.3. Even though some experiment results do vary across tasks as you suggested, it is more important that we have identified **a number of consistent and applicable findings that generalize across various parameter settings, tasks, and models**, including:
>
> - [line 370] PERFT-R generally is the best-performing strategy, and [line 414 & Figure 5] its performance is more sensitive to the overall expert count rather than the activated.
>     - As you pointed out, in line 418 we mentioned that “the optimal configuration appears to be task-dependent”, we were actually referring that to this finding. We believe this can serve as an empirical guide for those who want to design their own routed fine-tuning of MoE, where they can focus more on scaling overall expert numbers rather than exploring all design choices.
> - [line 369-374] Both PERFT-R and PERFT-E can outperform the qvLoRA baseline by significant margins and this holds across different models and parameter settings.
> - [line 377] PERFT-R and PERFT-E generally benefit from scaling bottleneck size within a certain range, whereas PERFT-S and PERFT-D don’t.
> - [line 464] Even fully-activated PERFT-R (TopN/N) can still improve the performance with its weight distribution among PEFT experts.
> - [Placement of Adaptation Modules for PERFT-E, line 511] When the total number of PEFT experts increases, PERFT-E’s stability becomes more preferred to PERFT-R’s flexibility.
>
> We hope to provide the research community with these applicable empirical findings we observed. We’ll rearrange these key findings as bolded paragraph titles to help better demonstrating our observations in the next revised version.
>
> ### Weakness 3 about Design of Figures:
>
> Thank you for pointing this out. We’ll try our best to improve Figure 4 with legends per your suggestions. Since there are too many important results to present and paper length constraints, to help reader interpret these results, a detailed breakdown of different TopK/M in PERFT-R has already been provided in Figure 5, and more results in Figure 7 & 8 in Appendix C.
>
>
> [1] He, J., Zhou, C., Ma, X., Berg-Kirkpatrick, T., & Neubig, G. (2021). Towards a unified view of parameter-efficient transfer learning. *arXiv preprint arXiv:2110.04366*.

---

> > ### Comment · Reviewer_T3WA · 2024-11-27
> >
> > Thanks for the response. I am still not convinced by the arguments provided here:  1. the unification in the draft is very straightforward and being "the first" on something like this is not novel; 2. "a number of consistent and applicable findings" are quite task/dataset dependent and I am not sure how applicable to new task/dataset. Thus, I am leaning to keep my original rating.

---

> > > ### Author Response · Authors · 2024-12-01
> > >
> > > Dear Reviewer T3WA,
> > >
> > > Thank you for your response! We appreciate it and would like to further clarify some points in our revised paper to help address your concerns.
> > >
> > > ### About novelty:
> > >
> > > The novelty of our work actually extends far beyond just being the first unified framework for PEFT in MoE. **We systematically explore possible designs for PEFT for MoE, and we especially discover the previously underexplored dynamics between vectors in routers for PEFT and FFN experts.**
> > >
> > > 1. **The dynamics between vectors in FFN experts, FFN expert router and PEFT expert router (visualized in Figure 3) provides a novel and intriguing understanding of how the vectors in the router for PEFT experts can interact with the vectors in the original router and the FFN experts** [line 209-238]. This brings the count of possible routing patterns to combinatorial numbers, which create much more exploration of hidden state patterns to be stored as vectors in routers. This insight suggests that **routing mechanisms can be leveraged not just for FFN expert selection, but also as a important integral part of the PEFT process itself, an insight that motivated our design and could also have significant implications for future MoE and PEFT research.**
> > > 2. We provide a **systematic study of how the PEFT modules can interact with MoE’s routing mechanisms**. We break down the detailed designs within PEFT and MoE, re-frame the possible solutions under functional and compositional strategies, and define a complete set of design dimensions along which different methods can vary. Any future method in this research direction which operates along these dimensions identified in our framework should benefit from our practical results in Section 4, which helps explore these design dimensions.
> > >
> > > ### About the applicability of our findings:
> > > Thank you for your concerns about applicability of our findings. We’d like to point out that **in our revised version, we reorganized our core results to better demonstrate our observations and findings that consistently generalize across all settings, domains, and models we’ve experimented.**
> > >
> > > We would also like to further clarify the previous misunderstanding about task-dependent configurations, which refers only for the sensitiveness of configurations under our observation 3 (see below) and calls for need of hyperparameter tuning in this case, instead of addressing all our results.
> > >
> > > In our [Section 4.2 Experiment Results], it is consistently observed that
> > >
> > > - **Observation 1. PERFT-R emerges as the best strategy** in terms of being able to achieve the best performance with extreme activated parameter efficiency.
> > > - **Observation 2. PERFT-R and PERFT-E generally benefit from scaling up.** For each configuration, the optimal settings can be found via scaling along the activated parameter efficiency.
> > > - **Observation 3 (3 to 2). PERFT-R is more sensitive to the overall number of PEFT experts** rather than the activated. Changing the total number of PEFT experts leads to more significant variation in performance, over changing the activated number of PEFT experts.
> > >
> > > We further analyze these observations, and summarize in [Section 4.3 Result Analyses] that:
> > >
> > > - **Finding 1: Routing is (very!) important in scaling the number of PEFT experts.** Without routing for PEFT experts leads to severe degradation of performance.
> > > - **Finding 2: Routing contributes more from its weight distribution, rather than sparse activation** (expert selection). Even PERFT-R with no sparsity can perform well, and sometimes even better.
> > > - **Finding 3: With more PEFT experts, PERFT-E can become favored over PERFT-R**, as PERFT-E sacrifices the flexibility (learning a independent routing for PEFT experts) for better stability (using the original routing pattern well-pretrained for FFN experts).
> > >
> > > **These observations and analyses are not merely based on the selected results presented in Section 4, but also from the comprehensive results in tables (Table 4,5,6,7,8) provided in Appendix C, with >160 PEFT settings being independently fine-tuned and >1100 evaluation scores reported across 14 benchmark datasets, which are enough for drawing to our conclusion.** All the aforementioned empirical findings are able to generalize across all our experiments, and are directly applicable for the community in their designing similar PEFT and MoE approaches.
> > >
> > > We hope these clarifications can help address your concerns and better demonstrate the value of our contributions. Thank you for your time and effort!

---

> ### Comment · Reviewer_T3WA · 2024-12-02
>
> Thanks for the response. I am still not convinced by these arguments. There is no doubt that the authors conducted lots of experiments, but I did not find much insights from them. For example, some of the observations and findings are contrary to each other: "Observation 1. PERFT-R emerges as the best strategy", "Observation 3 (3 to 2). PERFT-R is more sensitive to the overall number of PEFT experts", and "Finding 3: With more PEFT experts, PERFT-E can become favored over PERFT-R". What's the general idea/insight behind these results? If I want to use PERFT for a new task with a new model, what should I do?

---

> > ### Author Response · Authors · 2024-12-02
> >
> > Thanks for the response! We’d like to point out that our results are not contradictory, but actually forming a set of design guidelines following a logical progression:
> > (Here we denote Observations as O1-O3 and Findings F1-F3.)
> >
> > Generally, PERFT-R is the best overall strategy and PERFT-E is the second-best [O1]. Both of these 2 requires searching the scale of hyperparameters to achieve optimal performance [O2], especially adjusting the total number of PEFT experts has more impact than tuning activation ratios [O3]. Based on these, we summarize our conclusions about routers [F1 & F2], and reveal that only for the specific edge case when scaling to very large number of PEFT experts, PERFT-E may become more favorable due to its better stability [F3].
> >
> > For implementation on a new task/model, we recommend following our results, that start with PERFT-R as the default, and first search for the optimal total number of PEFT experts, and then fine-tune the activation ratio if needed. If the scaling results in a very large number of PEFT experts (approaching the number of FFN experts) in PERFT-R, consider switch to PERFT-E and give it a try.
> >
> > Thank you for raising up this concern. We’ll include a more detailed guideline in our appendices to help the community apply our findings more effectively.

---

> > > ### Comment · Reviewer_T3WA · 2024-12-02
> > >
> > > Thanks for the reply. The provided trial and error approach by overfitting to the specific test data does not sound a solid method. Thus, even though there are lots of experiments conducted, I am sticking to my original rating of 5 due to lack of insights.

---

### Author Response · Authors · 2024-11-27
**Official Comment for our Revised Paper**

Dear Reviewers,

We sincerely appreciate your thorough and constructive feedback on our manuscript. We have carefully considered all your comments and suggestions, and made revisions to strengthen the paper. All major adjustments are highlighted in yellow in the revised PDF. We outline some specific changes made in this version:

- In Abstract and Section 1. Introduction:
    1. We further clarified our research focus on MoE LLMs, and refined the current research landscape in MoE and PEFT;
    2. We better positioned the motivation of our work, and revised our articulation used for the unified framework and PERFT family.
- In Section 4. Experiments and Analyses:
    1. We add a legend to Figure 4 to help demonstrate our results;
    2. We reorganized Section 4.2 and 4.3 with our key observations directly as titles, to help making our findings more accessible.

We believe these revisions can help strengthen our paper and better demonstrate our core motivations and contributions. In summary,

1. Our work represents the first systematic study of PEFT for MoE LLMs. The unified framework we introduce comprehensively covers all possible design dimensions for applying PEFT to MoE models. We also reveal novel insights about the dynamics between key memory vectors in experts and expert vectors in routers - an interaction that had never been analyzed before in MoE and PEFT research.
2. To validate our framework, we propose the PERFT family and conducted extensive experiments at a scale previously unseen in MoE fine-tuning research. Through these experiments, we identify applicable findings for the research community that generalize across different settings, tasks and models.

We hope that our paper would contribute meaningful progress for the community in understanding how to efficiently adapt MoE models for downstream tasks.

Best regards,
The Authors

---

### Meta-Review · Area_Chair_zbsR · 2024-12-13

**Metareview:**

This paper presents a unified framework for incorporate parameter efficient fine tuning (PEFT) techniques into mixture of experts (MoE) LLMs. Reviewers identified the framework as being sound, and appreciated the experiments that were ran. Reviewers were concerned that this paper simply combined existing techniques and did not present any real insights.

The paper has scores of 8,5,3. When I asked the borderline reviewer (5) to move towards a more decisive opinion they said that they were leaning more towards a (weak) rejection. On reading the reviews, and the (long) exchange between the authors and Reviewers hrjf, there appear to be several unaddressed concerns. Indeed, not many strengths are actually identified.  As the scores are negative leaning, and the actual research contributions of this work are unclear (beyond it being a combination of existing work) I am inclined towards reject.

**Additional Comments On Reviewer Discussion:**

Reviewer T3WA stuck with their score after an exchange as they remained unconvinced that this work produced insights that generalised to new tasks. After an extensive exchange, reviewer hrjf felt that a lot of their questions were unanswered by the authors. Reviewer SUvZ on the other hand, raised their score, saying they were happy that the authors had addressed their doubts (although did not provide much further detail). As there are still lots of concerns that remain unaddressed, and a general lack of any strengths of the work being lauded I gave more weight to the well-explained rejects, as opposed to the more terse accept.

---

### Decision · Program_Chairs · 2025-01-22

Reject